

Hydrological controls on DOC:nitrate resource stoichiometry in a lowland, agricultural
catchment, southern UK.
Catherine M. Heppell[1]*, Andrew Binley[2], Mark Trimmer[3],Tegan Darch[1,2], Ashley
Jones[1,2], Ed Malone[1,2], Adrian L. Collins[4], Penny J. Johnes [5], Jim E. Freer[5] and
Charlotte E.M. Lloyd[6].
[1] School of Geography, Queen Mary University of London, Mile End Road, London
E1 4NS, UK.
[2] Lancaster Environment Centre, Lancaster University, Lancaster, LA1 4YQ, UK.
[3] School of Biological and Chemical Sciences, Queen Mary University of London,
Mile End Road, London E1 4NS, UK.
[4] Sustainable Soils and Grassland Systems Department, Rothamsted Research,
North Wyke, Okehampton, EX20 2SB, UK.
[5] School of Geographical Sciences, University of Bristol, University Road, Bristol
BS8 1SS, UK.
[6] School of Chemistry, University of Bristol, Cantock's Close, Bristol BS8 1TS, UK.
*Correspondence to Catherine M. Heppell (c.m.heppell@qmul.ac.uk)
**Abstract**
The role that hydrology plays in governing the interactions between dissolved
organic carbon (DOC) and nitrogen in rivers draining lowland, agricultural
landscapes is currently poorly understood, yet important to assess given the
potential changes to production and delivery of DOC and nitrate arising from climate
change. We measured DOC and nitrate concentrations in river water of six reaches
of the lowland River Hampshire Avon (Wiltshire, southern UK) in order to quantify the
relationship between Baseflow Index (BFI) and DOC:nitrate molar ratios across



contrasting geologies (Chalk, Greensand and clay). We found a significant positive
relationship between nitrate and Baseflow Index (p<0.0001), and a significant
negative relationship between DOC and Baseflow Index (p<0.0001), resulting in a
non-linear negative correlation between DOC:nitrate molar ratio and Baseflow Index.
In the Hampshire Avon, headwater reaches which are underlain by clay and
characterised by a more flashy hydrological regime are associated with DOC:nitrate
ratios > 5 throughout the year, whilst groundwater-dominated reaches underlain by
Chalk, with a high Baseflow Index have DOC:nitrate ratios in surface waters that are
an order of magnitude lower (< 0.5). Our analysis also reveals significant seasonal
variations in DOC:nitrate transport and highlights critical periods of nitrate export
(e.g. winter storm events in sub-catchments underlain by Chalk and Greensand, and
autumn events in drained, clay sub-catchments) when DOC:nitrate molar ratios are
low, suggesting low potential for in-stream uptake of inorganic forms of nitrogen.
Future work should determine whether the results reported here are transferable to
other agricultural, lowland catchments, and seek to understand the generalised
hydrological controls on the availability of DOC transported through such
landscapes.

## 1 Introduction

As we enter the Anthropocene, the increase in  nitrogen concentrations in the natural
environment, arising from the combined effects of agricultural intensification and
fossil fuel use, is causing pressing environmental problems (Vitousek et al., 1997;
Carpenter et al., 1998; Galloway and Cowling, 2002; Rabalais, 2002). An increase in
concentrations and loads of nitrate in freshwater environments is one such issue
arising from diffuse agricultural pollution, often correlated with the eutrophication of
coastal areas (Billen et al., 2011; Houses of Parliament, 2014; Howarth et al., 2012;
Vitousek et al., 2009; Withers et al., 2014). Furthermore, in permeable geologies,
responses to land management initiatives targeted at reducing nitrate loading are
delayed due to long water residence times, with little effect seen in some
groundwater-fed catchments over decadal timescales (Howden et al., 2011;
Tesoriero et al., 2013;  Wang et al., 2012; Wang et al., 2013; Wang et al., 2016). In
the United States, a legacy of accumulated nitrate in heavily managed, agricultural



catchments has been associated with temporal invariance of annual flow-weighted
concentration (a biogeochemical export regime termed chemostatic) irrespective of
the permeability of the geology and soil type (Basu et al., 2010). These managed
catchments are considered to be transport limited with regards to nitrate; meaning
that solute export is controlled predominantly by hydrology rather than
biogeochemistry (Basu et al., 2011). Thus changing climate, with important, potential
implications for rainfall patterns and hydrochemical responses in rivers, is adding a
new urgency to understanding and managing the issue of excess nitrate in our
agricultural-dominated landscapes (Howarth et al., 2011). In the UK, there is concern
that warmer, drier summers and wetter winters may lead to increased nitrate export
from lowland catchments (Whitehead et al., 2009), one scenario being an increased
accumulation of nitrate in soils by mineralisation in hot, dry summers followed by
flushing of nitrate from soils during autumn at the end of the drought (Whitehead et
al., 2006) especially in conjunction with first-flush responses (Jiang et al., 2010;
Yang et al., 2015; Orr et al., 2016). However, considerable uncertainty exists around
current predictions (Heathwaite, 2010); and policymakers lack results from studies at
appropriate temporal and spatial scales for confident decision-making (Watts et al.,

78  2015).

Over the last decade, there has also been an increasing awareness of the
significance of the transport and transformation of carbon in fluvial systems within
the overall conceptualisation of the global carbon cycle; and freshwaters are now
recognised as critical contributors to global carbon fluxes (Dagg et al., 2004; Beusen
et al., 2005; Battin et al., 2009). In addition, there is an increased understanding that
establishing the factors that control water-borne carbon fluxes is key to predicting the
likely implications of climate change for patterns and magnitude of organic carbon
transport through freshwaters (Aitkenhead and McDowell, 2000). Although dissolved
organic carbon (DOC) plays a crucial role in stream ecology (influencing processes
such as nutrient uptake and the balance between heterotrophy and autotrophy) our
understanding of terrestrial-to-aquatic transfers, aquatic processing of DOC and its
character in lowland, agricultural streams is incomplete (Stanley et al., 2012; Yates
et al., 2016, Aubert et al, 2013) as much of the effort in this area has been focused
on forested catchments, boreal peatlands and/or upland landscapes with significant
wetland cover (Frost et al., 2006; Ågren et al., 2007).





Macronutrients are not cycled in isolation, and important ecological consequences
arise from their interplay (Dodds et al., 2004); a key focus of current research is on
the linkage between essential nutrients such as carbon (C) and nitrogen (N).
Although these elements exist in many forms in river systems, the most abundant
biologically-available form of the compounds in lowland, intensively farmed
catchments are likely to be DOC and nitrate (Taylor and Townsend, 2010) with
nitrate typically contributing > 70% of the total dissolved N species (Durand et al.,
2011). The speciation of N in lowland agricultural catchments in Europe has been
reported previously (see Durand et al (2011), including in one of the sub-catchments
(River Wylye) that is a component of this study (Yates and Johnes, 2013; Yates et
al., 2016), but without comparison to the simultaneous behaviour of DOC.  This
paper therefore focuses on both nitrate and DOC, as the availability of DOC in a
stream ecosystem may influence both the quantity and speciation of nitrogen
exported downstream (Goodale et al., 2005;  Bernhardt and Likens, 2002;
Grebliunas and Perry, 2016). Taylor and Townsend (2010) synthesised global
datasets for DOC:nitrate ratios from groundwater to the open ocean, and
hypothesised that an observed threshold ratio of around four was indicative of the
shift in carbon to nitrogen limitation in rivers representative of the stoichiometric
demands of microbial anabolism. Taylor and Townsend (2010) suggested that, at
low DOC:nitrate ratios, the extent of nitrate accrual in global waters may be restricted
by the rapid conversion of nitrate to nitrogen ($N_2$) gas via denitrification, whereas at
high DOC:nitrate ratios heterotrophic nitrogen assimilation may strongly reduce in-
stream nitrate concentrations. Whole-stream nutrient additions to rivers
characterised by varying land use (using the 'Tracer Additions as Spiralling Curve
Characterisation' methodology) have provided experimental evidence that
DOC:nitrate ratios are strongly positively correlated with the rate of whole stream
nitrate removal (see results from Mulholland et al. (2015) presented in Figure 7 of
Rodríguez-Cardona et al. (2016)), although such experiments cannot distinguish
between nitrate removal via assimilation and/or denitrification mechanisms. In
summary, there is a need to understand whether monitoring DOC:nitrate ratios in
rivers could prove a useful component of a toolkit  for adaptive nitrate management
of river catchments in response to, for example land use or climate change.





Across the UK, nitrate concentrations in rivers are controlled by land use, relative
contribution of baseflow to streamflow and effective rainfall (Davies and Neal, 2007).
The problem of excessive nitrate export and high nitrate concentrations arising from
agricultural practice are most pressing in lowland catchments, such as the
Hampshire Avon, the focus of this study. In agricultural catchments, possible
management solutions need to be targeted to suitable scales of implementation,
such as farms and sub-catchments (Collins et al., 2016; Johnes et al., 2007). Thus,
our work considers different tributaries of the Hampshire Avon characterised by three
geologies (Chalk, Greensand and clay) with a range of groundwater influence. Within
this catchment, we predict that annual average nitrate concentrations will be
positively correlated with Baseflow Index following the findings of Davies & Neal
(2007) and we seek to establish any relationship between Baseflow Index and DOC,
and between Baseflow Index and DOC:nitrate ratio.

Controls on riverine DOC and nitrate arise from a combination of terrestrial
accumulation, transfer to the river and in-stream transformations (Stanley et al.,
2012).  The transfer of DOC and nitrate from terrestrial sources to the channel by
hydrological mechanisms results in changing relationships between concentration
and river discharge, often described by a power function ($C=AQ^b$), which can exhibit
marked intra-annual dynamics (Oeurng et al., 2011; Morel et al., 2009;  Basu et al.,
2010; Outram et al., 2014). Therefore, integrated annual measurements risk masking
important seasonal patterns in terrestrial-to-aquatic transfers and export of DOC and
nitrate, arising from variations in hydrological pathways throughout the year, such as
the interplay between groundwater and shallower lateral flows due to wetting up of
upper soil horizons in response to autumn rain (Prior and Johnes, 2002; Sandford et
al., 2013; Outram et al., 2014; Yates and Johnes, 2013). Such intra-annual variations
in solute chemistry have been termed the 'hydrochemical signature' of the catchment
(Aubert et al., 2013). This hydrochemical signature is especially important to
consider across an agricultural landscape characterised by a wide range of Baseflow
Index within which we might hypothesise that groundwater dominated areas will
exhibit a stable, more damped, hydrochemical response throughout the year,
whereas sub-catchments of low Baseflow Index might exhibit a wider range of nitrate
and DOC concentration arising from varying contributions of rapid hydrological




pathways (i.e. quickflow). Thus, here we aim to develop a spatio-temporal
understanding of the processes controlling loading of DOC and nitrate to a
catchment, which is essential for understanding and managing their combined
ecological impact. Furthermore, as our study took place during a period of drought
and subsequent flooding in the UK, a focus on seasonality may help to identify any
critical periods of nutrient export under future climate change scenarios of drier
summers and wetter winters.
To summarise, our research objectives were as follows:
(i) To quantify the relationship between nitrate, DOC and DOC:nitrate molar
ratio with Baseflow Index for six sub-catchments of contrasting geology in
the Hampshire Avon.
(ii) To assess the intra-annual variations in contributions of groundwater and
quickflow to streamflow across three sub-catchments representing high,
intermediate and low Baseflow Index, and; establish the extent to which
nitrate and DOC transport in the catchment arises from the interplay
between groundwater and quickflow components.
(iii) To quantify spatio-temporal differences in DOC:nitrate ratios in this
agricultural landscape and assess the potential implications of these
variations for future nitrogen management.

**2 Materials and methods**
**2.1 Site description**
The research was undertaken at six river reaches in the Hampshire Avon
upstream of Salisbury (Wiltshire, UK), representing sub-catchments of contrasting
geology (clay, Greensand and Chalk), and a gradient of Baseflow Index (Figure 1;
Table 1). The majority of the upper catchment of the Hampshire Avon (draining c.
1390 km$^2$ in total), is dominated by the Cretaceous Chalk geology, and the
hydrogeological properties of these geological units are described in detail in Allen et
al., (2014). Sites CW on the river Wylye and CE on the river Ebble are river reaches
characterised by high baseflow indices (>0.9) where Chalk provides the main source
of groundwater (Allen et al., 2014). In the north and west of the Hampshire Avon
catchment there are also significant groundwater contributions from geological



formations of Upper Greensand which comprise fine-grained glauconitic sands and
sandstones (Bristow et al., 1999). The sub-catchments of sites GN on the river
Nadder in the west of the catchment, and GA in the north of the catchment, both
comprised c. 50 % Upper Greensand by area with baseflow indices of 0.695 and
0.861, respectively. The two sites characterised by the lowest baseflow indices, sites
AS (0.372) and AP (0.234), are located in the sub-catchment of the river Sem
underlain by impermeable Late Jurassic Kimmeridge Clay (usually a non-aquifer)
and thin interbedded limestone from which limited groundwater flow may occur (Allen
et al., 2014). Agricultural land use dominates the Hampshire Avon catchment with
arable farming including horticulture comprising 42% of land use, and improved
grassland for dairy and beef production covering 23% of the catchment (Natural
England, 2016). The distribution of arable and livestock farming varies with sub-
catchment; improved grassland dominates in the clay catchment of the river Sem
(AS and AP), where it supports intensive dairy production, whilst arable agriculture
represents circa 50% of land use at the chalk sites (CW and CE), with sheep grazing
and intensive pig production as minority land uses (Table 1).

**2.2 Field instrumentation**

210         Sites AS, GA, GN and CE were instrumented for two years from June 2013

until June 2015. Stream stage was measured using pressure transducers (HOBO
U20-001-01, Onset Corporation, USA at AS, GA and GN; Levelogger Edge, Solinst,
Canada at CE) in a perforated stilling well, logging at 15-mins intervals. Regular
(fortnightly when possible) manual measurements of discharge by the velocity-area
method enabled construction of stage-discharge relationships for each site.
Discharge values used in the analysis were scaled to mm day$^{-1}$, using an assumed
catchment area defined by the topographic divide for that point in the stream
network. Rainfall was measured at 15-mins intervals at AS, GA and CE using a
tipping bucket raingauge (674, Teledyne ISCO,USA) in order to calculate daily
rainfall totals (mm d$^{-1}$) for the study period. Details of exact locations of hydrological
measurements can be found in Heppell et al. (2016a, 2016b).





Temperature, pH, temperature, dissolved oxygen (optical) and electrical conductivity
of river water were logged in-situ at 30 mins intervals using a Water Quality
Multiprobe (Manta 2, Eureka Water Probes, USA). An automatic water sampler
(6712, Teledyne ISCO, USA) collected water samples from the river every 48-hrs
from June 2013 to June 2014 for analysis of water chemistry, and samples were
collected fortnightly. Therefore, field and laboratory tests were undertaken to ensure
that sample degradation over this time period was negligible. Furthermore, MilliQ
water was decanted into sample bottles in the field to create field blanks to ensure
no sample contamination occurred during transportation between the field and
laboratory. Three riparian piezometers (screen depth installed in the soil C horizon,
typically circa. 2 m depth) with porewater sampling tubes at screen depth were
installed in the banks at each site in summer 2013 to enable measurements of
riparian hydraulic head and porewater samples to be collected for chemical analysis.
Hydraulic head was measured using pressure transducers (HOBO U20-001-01,
Onset Corporation, USA at AS, GA and GN; Levelogger Edge, Solinst, Canada at
CE) validated with manual dips on a fortnightly basis. Porewater samples were
collected from sampling tubes on the riparian piezometers every two months from
February 2014 to June 2016 using a syringe and tygon tubing. Samples were then
filtered to 0.45 $\mu$m in the field.

Sites AP and CW were a component of the Demonstration Test Catchment network
(McGonigle et al., 2014; Outram et al., 2014). At AP, stream discharge was
measured using a Mace Flow Pro to record paired stage height and velocity
measurements at 15-min temporal resolution to which the velocity-area method was
applied (Lloyd et al., 2016a,b). The Mace Flow Pro measurements were taken within
a concrete section which meant that the cross-sectional area was stable. However,
during high flow events, the stage height overtops the concrete structure and out of
bank flows occur. In these cases, a weir equation was implemented to account for
the additional water flowing over the concrete section:

$Q_i = C_d b H_i^{1:5}$



where: $Q_i$ is the discharge at time point $_i$ (m$^3$ s$^{-1}$), $C_d$ is the dimensionless coefficient
of discharge, b is the weir crest breadth (m) and $H_i$ is the stage height (m) above the
bridge at time point $_i$. $C_d$ was set at 2.7 based on typical values from published
literature (Brater and King, 1976). Discharge data for CW were obtained from the
Environment Agency Gauging Station (Gauge number 43,806), which provided 15-
min resolution stage height data using a Thistle 24R Incremental Shaft Encoder with
a float and counterweight. For periods of modular flow, these data were used in
conjunction with a stage-discharge curve to calculate discharge (ISO 1100-2, 2010).
However, during non-modular flow periods, the stage heights are used alongside 15-
min velocity measurements from a second ultrasonic gauge to calculate discharge
using the velocity-area method (ISO 1088, 2007). At both sites daily river water
samples were collected using automatic water samplers (3700, Teledyne ISCO,
USA).

**2.3 Laboratory analysis**
On return to the laboratory a sub-sample of river water from sites AS, GA, GN and
CE was filtered at 0.45 µm for analysis of nitrate and DOC. Nitrate concentrations
were analysed using ion exchange chromatography (Dionex-ICS2500). The limits of
detection (LOD) and precision were 8 µmol L$^{-1}$ ± 7 %. These samples were then
prepared for DOC analysis by acidification to pH < 2 with HCl and then analysis by
thermal oxidation (Skalar) using the non-purgeable organic carbon (NPOC) method.
The LOD of the DOC analysis was 42 µmol L$^{-1}$ with precision of ± 12 %. Accuracy
was ensured by analysis of certified reference material (SPS-SW2 and TOIC4M14F1
for nitrate and DOC respectively) with each instrument run. Porewater samples from
all sites were analysed using the same methods as for the surface water from AS,
GA, GN and CE.
River samples collected from sites AP and CW were filtered then analysed for nitrate
using a Skalar San++ multi-channel continuous flow autoanalyser. This analysis was
based on the hydrazine-copper reduction method producing an azo dye measured
colorimetrically at 540 nm. DOC was analysed as non-purgeable organic carbon by
coupled high temperature catalytic oxidation using a Shimadzu TOC-L series





analyser. For further details on sample collection and analysis at AP and CW sites
see Yates et al., 2016.

## 2.4 Data analysis

Baseflow Index (BFI) for each site was calculated using the hydrograph separation
procedure outlined in Gustard et al. (1982). Hydrographs with high BFI show
relatively smooth characteristics and are indicative of major aquifers where water
(and consequently solute) residence time in permeable bedrock will be of the order
of decades whereas a low BFI is characterised by a flashy hydrograph, with steep
recession curves, and is indicative of a generally shorter residence time in the
catchment before water reaches the stream channel, with quickflow comprising
shallow, lateral preferential and overland pathways predominant during storm
events. Soil moisture deficit (SMD) is defined as the amount of water (in mm) which
would have to be added to the soil in order to bring it back to field capacity. SMD
values were obtained from the UK Meteorological Office for MORECS square 169
(4000 east, 1400 north) for a medium textured soil type with predominantly grass
cover.
In order to quantify the relationships between nitrate, DOC and DOC:nitrate molar
ratio with Baseflow Index, and to understand how any relationship varied intra-
annually a linear mixed effects modelling approach was used. Linear mixed effects
models account for missing data, which is a common issue associated with long-
term field datasets, and the inclusion of repeated measures in the analysis
(Blackwell et al., 2006). The 'lmer' function in R (R Core Team, 2016) package lme4
(Bates, Maechelr & Bolker, 2015) was used to perform a linear mixed effects
analysis of the relationship between BFI as the independent measure, and either
nitrate concentration, DOC concentration or DOC:nitrate molar ratios as the
dependent variable.  The nitrate and DOC concentration of river water recorded at
each site over the same time period (i.e. from samples collected at simultaneous 48-
hr time intervals from June 2013 until June 2014) was used in the analysis. BFI was
entered as a fixed effect. We accounted for the influence of repeated measures by
including time (Julian Day) as a random intercept and slope in the model. The 'lme'
function in R package 'nlme' (Pinheiro et al., 2016) was used to fit a linear mixed





effects model to porewater data to investigate differences in nitrate and DOC
concentrations between CE and CW (the Chalk sites) and all the other sites (AS, AP,
GA and GN).

For the purposes of considering the relationship between BFI and nitrate
concentrations in the wider Hampshire Avon catchment, nitrate concentrations in
river water samples collected between June 2013 and June 2014 were obtained
from the Environment Agency Harmonised Monitoring Scheme (HMS) Records.
Average annual nitrate concentration was calculated for each site, but those with
less than 12 samples in the 12-month period were removed from the analysis.
Baseflow index for each Environment Agency site was estimated using the Flood
Estimation Handbook which uses the Hydrology of Soil Types (Boorman et al., 1995)
methodology because there is not a gauging station at every location. Baseflow
indices derived in this manner are referred to as BFIHOST to distinguish them from
BFI values derived using our own discharge data.

Annual loads of nitrate and DOC for sites AS, GA and CE were calculated as kg ha$^{-1}$
by integrating paired concentration and discharge data collected on a 48-hr basis
from June 2013 to June 2014. Any missing solute data (maximum gap of 10 days
due to equipment failure) were infilled using seasonal concentration-discharge
relationships derived for each site. Seasonal loads are expressed as a percentage of
total annual load for each site.

**3 Results**
**3.1 Rainfall and soil moisture deficit during the study period**
The first year of study (June 2013-2014), on which these results are focused, was
characterised by pronounced cycles of soil wetting and drying due to alternating
periods of unusually wet and dry weather (Figure 2). Due to a combination of lower-
than-average rainfall (c. 50% of 1910-2015 long term average for the region) and
high temperatures (>28°C for a 10-12 day period in July) over the summer of 2013,



soil moisture deficit (SMD) reached a maximum of 140 mm for a 4 week period in
August and September 2013. A period of unsettled weather in October and
November 2013 (224 mm rainfall in total) reduced the SMD to 0 mm. After a brief
return to dry, settled conditions, a series of deep Atlantic low pressure systems
brought a prolonged period of heavy rain to the entire Hampshire Avon catchment.
161 mm rain fell in December 2013 (190% of the 1961-1990 long term average), with
a maximum daily rainfall total of 58 mm on 23 December, followed by a further
monthly total of 205 mm and 148 mm in January and February 2014, 261 and 259 %
of the long-term averages, respectively. January 2014, in particular, was the equal-
wettest on record since 1910.  SMD and rainfall patterns in 2014 were not as
extreme as those in 2013, returning to monthly values that were much closer to the
long term averages. SMD reached peak values of 129 mm by the end of the summer
in early October 2014, and autumn rainfall during October and November caused
wetting up of the soil to reduce SMD to 0 mm by mid-November 2014. By March
2015, warmer weather, combined with lower-than-average rainfall (< 50% of long
term average) caused SMD to steadily increase until the end of the study period in
June 2015.

### 3.2 BFI and nutrients

Nitrate concentration in surface water of our sub-catchments is significantly positively
correlated with BFI (r=0.749, p<0.001), whereas DOC concentration in our surface
water samples exhibits a significant negative correlation with BFI (r=-0.881, p<0.001)
(Figure 3a & 3b, Table 2). The linear mixed effects model analysis indicates that BFI
has a significant effect on nitrate ($\chi^2$(1)=19.348, p<0.0001) and DOC ($\chi^2$(1)=497.82,
p<0.0001) concentrations, with an increase in BFI of 0.5 leading to a difference in
average increase in surface water nitrate concentrations of 260 $\mu$mol L$^{-1}$ and a
reduction in DOC concentrations of 840 $\mu$mol L$^{-1}$ between the clay and Chalk sites.
Inclusion of time as a random effect (both slope and intercept) improved the model fit
for both nitrate and DOC, indicating that temporal dynamics associated with these
determinands are important to consider. The sites of lower BFI exhibit marked
variations in nitrate concentration in autumn and winter, which change the slope
(although not the overall direction) of the nitrate and BFI relationship, and highlight





380 the importance of seasonality. Overall, the respective increase in nitrate, and

381 decrease in DOC concentration with BFI, broadly reflects the patterns in

382 concentrations of DOC and nitrate in the riparian zones associated with each

383 geology. Nitrate concentrations in riparian porewaters were significantly higher in the

384 Chalk sites compared to the clay and Greensand sites ($F_{(1,146)}$=105, $p<0.0001$),

385 whereas DOC concentrations were significantly lower in the Chalk sites compared to

386 the others ($F_{(1,146)}$=38, $p<0.0001$). The relationship between DOC:nitrate molar ratio

387 and BFI is non-linear and can be best described by a power function

388 (DOC:nitrate)=0.453*$BFI^{-2.575}$, $r^2$=0.638, $p<0.001$, Figure 3c).

389

390 The relationship between nitrate and BFIHOST was tested for 17 additional sites

391 within the Hampshire Avon catchment using Environment Agency Harmonised

392 Monitoring Scheme data collected between June 2013 and June 2014. Figure 4a

393 shows that across the Hampshire Avon, there is a significant, positive, linear

394 relationship between nitrate and BFIHOST (r=0.951) with a regression model

395 indicating that BFIHOST accounts for 90.4% of the variation in nitrate concentration.

396 There is also a significant, positive correlation between nitrate concentration and %

397 arable land use (r=0.839, $p<0.001$). Although % arable and BFIHOST are positively

398 correlated (r=0.881), a tolerance value (a test for collinearity) of 0.224 indicates that

399 multiple linear regression can be used in this instance (Field, 2000). Multiple

400 regression shows, however, that BFIHOST alone produces the best model, with the

401 forced inclusion of % arable resulting in no significant improvement to the model fit

402 (Table 2).

403

404 **3.3 Contrasting hydrochemical signatures of three sites of low, intermediate**

405 **and high BFI**

406

407 From this point forward, data from three sites only are presented as illustrative of the

408 hydrochemical signatures from a range of baseflow indices across our three

409 geologies; Chalk (Site CE – high BFI), Greensand (Site GA – intermediate BFI) and

410 clay (Site AS – low BFI). There is a marked difference in the response of electrical




conductivity to discharge across the three sites (Figure 5a-c). At the chalk site, CE, a
maximum electrical conductivity of 0.570 mS cm$^{-1}$ is maintained across the full range
of recorded discharge. At the Greensand site, GA, electrical conductivity is
maintained at c. 0.650 mS cm$^{-1}$ until discharge exceeds 1 mm d$^{-1}$ and then a decline
in electrical conductivity with increasing discharge is observed. An examination of
electrical conductivity by season indicates that geogenic solute concentration was
lowest at the Greensand site during winter 2014, and concentrations were
comparable in spring, summer and autumn (Figure 5b). At the clay site, AS, there
are two different relationships between electrical conductivity and discharge; a
constant electrical conductivity of c. 0.520 mS cm$^{-1}$ is maintained at lower discharges
of 0.001 – 0.3 mm d$^{-1}$, whilst a log-linear decrease in electrical conductivity is
observed between 0.2 and 3.5 mm d$^{-1}$, and there is some overlap between the two
patterns of behaviour. Box-plots of electrical conductivity by season indicate highest
concentrations of geogenic solutes in summer, intermediate concentrations in
autumn and spring, and lowest concentrations in winter (Figure 5c).

Inter-site comparisons of the response of nitrate, DOC and DOC:nitrate molar ratio to
variations in discharge are illustrated in Figure 6. There is a significant, positive
correlation between log-nitrate and log-discharge for all sites, with the slope of the
regression relationship increasing with BFI (CE<GA<AS; Table 3). Visual
examination of the relationship between nitrate and discharge for AS and GA
suggests more than one trend is apparent and this is investigated in detail by
considering seasonality below. There is also a significant, positive correlation
between log-DOC and log-discharge, although in this case the slope of the
regression relationship increases in the following order: AS<CE<GA (Table 3).
However, again there is marked scatter in the relationship and this is investigated
further below. There is a similar significant, proportional increase in DOC:nitrate
molar ratio with increasing discharge at both CE and GA (slopes of 0.199 and 0.196,
respectively on a log-log basis, Table 3) whilst AS has a much weaker relationship,
exhibiting far greater scatter.





### 3.4 Seasonality of concentration-discharge relationships for three selected sites

Nitrate concentrations at the Chalk Site, CE, show little variation with season or discharge, whereas DOC concentrations appear to follow two trends; (i) a slight increase in DOC concentration with discharge in spring and winter; and (ii) elevated concentrations of DOC which are unrelated to discharge in spring (Figure 7a). Consequently, DOC:nitrate molar ratios remain low (<1) throughout the year (Table 4).

At the Greensand site, GA, both nitrate and DOC concentrations increase with discharge (irrespective of season) until a breakpoint is observed at 1.5 mm d$^{-1}$. At this point, during the winter storms of 2013-14, nitrate concentrations start to decline with increasing discharge whereas DOC concentrations drop to < 500 µmolL$^{-1}$ and a new, positive trend in increasing DOC with increased discharge is observed with a gentler slope (Figure 7b). As a consequence, the positive relationship between DOC:nitrate ratios and discharge also show a similar breakpoint, but the DOC:nitrate ratio remains below 3:1 throughout the year (Table 4).

At the clay site, AS, there are two trends in the concentration-discharge relationship for nitrate (Figure 7c). Concentrations are highest (200-400 µmol L$^{-1}$) during the autumn storms of intermediate discharge that followed the summer drought of 2013. The winter storms of 2014 are associated with highest discharge, but lower nitrate concentrations (c. 100 µmol L$^{-1}$). This contrasts with DOC which shows a plateau in concentration (c. 1000 µmol L$^{-1}$) with increasing discharge, irrespective of season. Nitrate and DOC concentrations were plotted against electrical conductivity to test whether nitrate and DOC arose from a linear combination of old and new water, but this was not the case (data not shown) suggesting that variations in supply and/or in-stream processing of these solutes occurs through the seasons. At AS, there are two observable trends in DOC:nitrate molar ratio: (i) highest and the greatest variability in DOC:nitrate ratios are observed during summer low flow conditions; (ii) there is an increase in DOC:nitrate ratios with discharge irrespective of season (Figure 7d).



Consequently, during autumn, values of DOC:nitrate ratios were generally equal to
or less than five, whilst values significantly greater than the threshold of four
observed by Taylor and Townsend (2010) predominated during the spring, summer
and winter (Table 4).

Over 50% of the annual DOC load was exported from our sub-catchments during
winter months, irrespective of geology. In the spring, 22-28% of the annual DOC load
was transported, with summer and autumn months together responsible for < 20% of
the total weight of DOC leaving each sub-catchment (Table 4). Winter was also an
important season for nitrate export with between 45 and 66% of the total annual
nitrate load being exported. Spring export of nitrate was important in both Chalk and
clay sub-catchments (c. 30% of annual load) and in the clay, autumn export of nitrate
was also of comparable magnitude to spring (Table 4).

**4 Discussion**
**4.1 Contrasting hydrological responses across a gradient of BFI**
Our six sites exhibit a range of BFI (0.207-0.905) indicating a gradient from river
water with 80-90% groundwater contribution to total flow in the chalk geology, 70-
80% groundwater contribution in the Greensand and only 20-55% groundwater
characteristic at the sites underlain by clay geology. Our calculation of BFI for the six
sites, based on our two-year discharge dataset, compared favourably with the BFI
estimated from HOST (Gustard et al., 1992).

BFI and logEC-logQ plots are useful complementary approaches to interpreting
hydrological and hydrochemical pathways operating in the sub-catchment associated
with each site. Electrical conductivity is an aggregated measure of geogenic solute
response in the sub-catchment, and provides an indication of relative contributions of
old groundwater (long residence time) and new (short residence time) water arising
from routes such as shallow throughflow, preferential pathways and overland flow to
the river. The study allowed the full range of flows at the sites to be sampled





because two extreme conditions in the UK were captured: the summer drought of
2013 and the extremely wet winter of 2013-2014. In the Chalk, the logEC-logQ plots
show groundwater (old water) dominance during the period of flooding, because
electrical conductivity is maintained through the entire range of flows, including at the
highest discharge approaching 10 mm d$^{-1}$. At the Greensand site, the sharp decline
in electrical conductivity at discharges >1.5 mm d$^{-1}$ provides evidence of dilution of
total dissolved solutes by new water, which occurs only during the wet winter of
2014. At the clay site, the data demonstrate that quickflow pathways, most likely
involving preferential delivery enabled by field drainage (both agricultural and army
camp drains from World War II) installed due to the risk of seasonal waterlogging on
the slowly permeable local clay soils (Denchworth and Wickham soil series), are
operational throughout autumn, winter and spring months. Under summer baseflow
conditions, the field drains are inactive and any river flow (almost negligible during
the summer drought of 2013) is provided by springs draining the aquifers of the
Upper Greensand and Wardour Formation (Allen et al., 2014), or direct discharges
from septic tanks, and drains connecting farm yards to the stream.

**4.2 Nitrate and DOC concentrations as a function of BFI**
Average annual nitrate concentrations in surface waters of the Hampshire Avon
catchment increase with increasing BFI. In a UK-wide study, Davies and Neal (2007)
used linear regression to consider how catchment characteristics control mean
nitrate concentrations in UK rivers. Nitrate concentrations were explained by land
use (% arable and % urban), topography (expressed as % upland), effective rainfall
(mm) and BFI. Therefore, on the basis of these prior national analyses, it would be
predicted that % arable and BFI would be the most important explanatory factors.
For the Hampshire Avon, stepwise regression analysis showed limited co-linearity
between BFI and % arable, and forced entry regression indicated that BFI was the
better explanatory variable for mean nitrate concentrations. In the UK, historical
fertiliser applications have led to elevated concentrations of nitrate in both Chalk and
Upper Greensand aquifers; currently in the range 500-645 $\mu$mol L$^{-1}$ (Defra, 2002;
Burt et al., 2011; Howden et al., 2011; Wang et al., 2016). Although the Chalk aquifer
of the Hampshire Avon has been designated as a Groundwater Nitrate Vulnerable



Zone (NVZ) under the EU Nitrate Directive (Directive 2000/60/EC), the time taken for
water to move from the soil surface, through the unsaturated zone to the aquifer can
result in a decadal scale time-lag between implementation of management practice
and any observed response in groundwater or river nitrate concentrations (Allen et
al., 2014; Wang et al., 2012). We observe an increase in nitrate load in baseflow with
increasing BFI (Chalk > Greensand > clay) in line with previous research by
Tesoriero et al (2013), and our riparian porewater samples indicate significantly
higher nitrate concentrations in the soil C horizon of the Chalk sites in comparison to
Greensand and clay sites. However, it is an over-simplification to suggest that the
gradient of annual average nitrate concentrations with BFI can be explained solely
by different ratios of nitrate-rich groundwater to relatively nitrate-poor quickflow
components of the hydrograph over an annual cycle. If this were the case, then
nitrate concentrations would be highly correlated with electrical conductivity, and
they are not. Instead, our analysis suggests that additional nitrogen transformation
processes, and exchange with other nitrogen species forms instream, driven by
seasonality and varying land use and management contribute to the observed
patterns that we see, and this is discussed below.

Our six sites provide evidence that average annual DOC concentrations decline with
increasing BFI in the Hampshire Avon catchment. Unfortunately, the Environment
Agency does not collect DOC data in the rivers of the Hampshire Avon region so we
cannot investigate the wider applicability of the DOC trend. Wetland area is often
cited as an important control on DOC concentrations in a catchment (Morel et al.,
2009), but our sub-catchments all comprise < 0.6% wetlands by area. Data from the
Environment Agency indicate that groundwater concentrations of DOC in the
catchment are generally < 83 $\mu$mol L$^{-1}$. Porewater samples from the grassland
riparian zone at each site show elevated DOC concentrations in comparison to
regional groundwater, and the Chalk sites (high BFI) have significantly lower DOC
concentrations in soil C horizons compared to the Greensand and clays, suggesting
that soil type and underlying geology could influence the concentration at which DOC
is delivered to the stream in these sub-catchments. Once again, DOC concentrations
in the surface water cannot be explained by a mix of old and new water alone, and
seasonality plays an important role in controlling the flux of DOC through river water.






### 4.3 Seasonal controls on nitrate and DOC export

The Chalk site (CE) is chemostatic with respect to total dissolved solutes i.e. overall, the concentration of geogenic solutes is maintained at higher discharge, so that discharge drives solute load and hence the export of solutes to the coast. This observation also holds for nitrate. It has been suggested that chemostatic behaviour for nutrients arises if sources accumulate in the landscape e.g. as legacy of nitrate management. Here nitrate has accumulated in groundwater (Wang et al., 2016) and it is the dominance of this old water under high discharge that gives rise to the chemostatic effect and transport-limited system. DOC is also transport rather than supply limited at this site, showing a slight increase in concentration with increasing discharge, and a more pronounced increase in spring which is not associated with a rise in discharge. In fact all three sites – on Chalk, Greensand and clay – have elevated DOC concentrations in spring, which could arise from mineralisation, leaching and export of DOC from catchment soils as soil temperatures rise (Aubert et al., 2013), and/or in-stream production.

584

At the Greensand site, there appears to be a threshold of discharge of c. 1.5 mm d$^{-1}$ in winter above which there is evidence of different hydrological flowpath(s) or sources of water to the river with lower electrical conductivity compared to other seasons. Riparian head is closely correlated with discharge and shows two distinct regions of linearity which converge at a discharge of between 1 and 1.5 mm d$^{-1}$. At this threshold, riparian head is at 60-80 cm below the ground surface suggesting that the water table is at the base of the soil C horizon. As the water rises up through the soil horizons during the winter, the electrical conductivity in the river water drops indicating a supply of new water from soil in the riparian zone and potentially from the surrounding fields. Conceptualisations of solute transport from other researchers include differing contributions from near stream riparian areas with rising and falling groundwater, arising from a combination of soil solute concentration and near-stream lateral water flux (Prior and Johnes, 2002; Seibert et al., 2009), and/or increased connectivity and fraction of active catchment contributing water, with emphasis on the lateral dimension (Basu et al., 2010). Above our threshold of 1.5 mm d$^{-1}$ the DOC



and nitrate concentrations in the river reflect a combination of groundwater
contribution and the depth-integrated mass flux of each solute from the soil A, B and
C horizons. The reason for a decline in nitrate concentrations in river water above
the threshold, whilst DOC concentrations increase, can be ascribed to the different
depth-distributions of nitrate and DOC pools in the soil. The extent of the lateral
connectivity between surrounding fields, the riparian zone and the river channel in
these low gradient, intermediate BFI systems is not well characterised, and should
be an area of further study.

Our two clay sub-catchments are dominated by artificially drained soils of the
Kimmeridge Clay Series, and the field under-drainage will be a major control on the
hydrological and hydrochemical response of the river. This is evident in the rapid fall
in electrical conductivity in response to rainfall events (data not shown) and in the
variation in electrical conductivity with season which arises from the mix of rapid (via
drainflow) and slow pathways of water during storm events, and suggests that the
drains operate through much of the year (spring, autumn and winter). Concentrations
of DOC in the surface waters of the two clay sites (167 – 2000 $\mu$mol L$^{-1}$) are
comparable to the range reported in drainage waters from permanent grassland in
South West England (Sandford et al., 2013). Increases in DOC concentrations in
drainage water during rainfall events have previously been explained as being due to
increased lateral flows through the upper soil horizons (Neff and Asner, 2001), which
are generally relatively carbon enriched compared to lower soil horizons. Here,
flushing of DOC from soil aggregates and subsurface micropores contributes to
rising concentrations during storm events (Jardine et al., 1990; Chittleborough et al.,
1992). Sandford et al. (2013) reported molar DOC:nitrate ratios of 18-25 at times of
highest DOC export in drainage water (which is at the upper end of our observations
for surface water of our clay catchment), and they also found that the molar
DOC:nitrate ratio increased with discharge. The comparability of results suggests
that our findings may have wider applicability to other catchments of mineral soils
dominated by drained grassland.



### 4.4 Ecological significance of temporal variations in DOC:nitrate ratio across a gradient of BFI

Here, we have shown that for our six tributaries of the Hampshire Avon, DOC:nitrate ratios are negatively correlated with BFI, but the relationship is non-linear. As far as we are aware, we are the first to demonstrate such a relationship, which, if more widely applicable to other lowland, agricultural catchments, might provide a useful means of predicting annual-averaged riverine nitrate and DOC concentrations.

The molar DOC:nitrate ratios fall in the lowest range recorded across multiple land use types in the US LINXII study (Mulholland et al., 2015), but vary over two orders of magnitude, suggesting order of magnitude variations in whole stream nitrate uptake velocity in river reaches across our contrasting geologies (0.05-0.4 mm min$^{-1}$; see Figure 7 in Rodríguez-Cardona et al., 2016). Nitrate uptake velocity is the vertical movement of nitrate to the riverbed measured using the whole stream 'Tracer Additions as Spiraling Curve Characterisation' method. The metric represents nitrate uptake efficiency, and can be interpreted as whole stream nitrate removal through, for example, denitrification and/or assimilatory processes, although the method does not allow for discrimination of these processes. On the basis of the relationship between DOC:nitrate and BFI demonstrated in this study, we can hypothesise that the clay sub-catchments are associated with higher whole stream nitrate removal than our greensand and chalk systems. Although we have no direct measurements of whole stream nitrate removal for these sites, we have measured in-situ rates of nitrate removal in the riverbed at these six sites using a modified push-pull technique (Jin et al., 2016), and the highest rates of nitrate removal were found at the two clay sites (see Table 4 in Jin et al., 2016). Whether DOC:nitrate ratios control nitrate removal may also depend on the net heterotrophic or autotrophic nature of our sub-catchments. In a net autotrophic reach, nitrate removal might correlate with physical factors such as light and temperature, which control photosynthetic activity, and hence the in-stream production of labile carbon which, in turn, is then tightly coupled to nitrate reduction. In contrast, in a net heterotrophic reach in our lowland, arable landscape, nitrate removal may depend on DOC:nitrate ratios driven by hydrological pathways delivering labile dissolved organic and inorganic carbon.

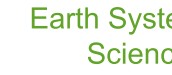 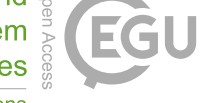

663

This study has revealed significant differences in the relationship between
DOC:nitrate and discharge dependent on both geology and seasonal effects. The
Chalk site exhibited little variation in DOC:nitrate with discharge due to the
dominance of groundwater contribution at both high and low flows. At the Greensand
site, there is a linear increase in DOC:nitrate with discharge irrespective of season.
However, during the elevated flows in the winter, when riparian and rain water
contributes increasingly to the discharge, causing a drop in Electrical Conductivity, a
sharp change in nitrate and DOC concentration is observed resulting in an overall
drop in DOC:nitrate during a time when > 66% of the total nitrate export occurs.  In
contrast at the clay site, lowest DOC:nitrate values and highest nitrate
concentrations are associated with autumn storms of intermediate discharge, which
export 26% of total annual nitrate load. These trends highlight contrasting seasons of
risk associated with high nitrate export in combination with low DOC:nitrate ratios at
the Greensand and clay sites. Our research gives added impetus to the need to
control autumn run-off from drained, grassland catchments supporting intensive
livestock farming and also suggests that periods of lateral flow from soils, and over-
bank flooding in areas of intermediate BFI, such as Greensand, may export a
significant proportion of the annual nitrate load with little opportunity for in-stream
nitrate processing or removal.

683

## 5 Conclusions

We have shown that the dynamism of hydrological pathways, here quantified using
BFI, is a controlling factor influencing both annual average DOC and nitrate
concentrations in heavily managed agricultural landscapes, and thus the extent to
which groundwater influence also affects DOC:nitrate ratios. In the Chalk sub-
catchment, a chemostatic nitrate response over the year is a consequence of the
dominance of nitrate-rich groundwater-flow, and nitrate export is transport-controlled.
Thus, under future climate change scenarios, periods of groundwater flooding such
as observed in winter 2013-4 will be critical periods of nitrate export with little
opportunity for in-stream nitrate processing and removal due to a combination of
short residence times, low water temperatures and low DOC:nitrate ratios (<0.5). In



sub-catchments of intermediate BFI, such as the Greensand sub-catchments in this
study, high winter flows, although arising from a mix of slow and rapid hydrological
pathways, may also be characterised by water with low DOC:nitrate ratios circa. 1,
suggesting that nitrate accrual rather than in-stream nitrate removal could be
promoted downstream.

Although heavily managed, the clay sub-catchment showed marked variation in
nitrate and DOC concentrations with discharge, driven by season; and little evidence
of chemostatic behaviour. In this sub-catchment there was a strong positive
relationship between DOC:nitrate ratio and discharge, and DOC concentrations were
generally higher than for our other landscape types. It seems that, at the landscape
scale, both quickflow and preferential flow through field drains may supply rivers with
a source of water conducive to promoting in-stream nutrient removal. Although care
should be taken to ensure that in such catchments, relatively high DOC
concentrations do not arise from pollutant sources with a high biochemical oxygen
demand (such as slurry), further work should focus on the sources and lability of
DOC from drained, grassland soils. At the landscape scale, it can be hypothesised
that the locations where water from impermeable sub-catchments meet water from
tributaries of lower BFI, may be hotspots of heterotrophic activity driven by upstream
supply of water with a high DOC:nitrate ratio. In this way, the spatial arrangement of
areas of contrasting BFI within a catchment may have important ecological and
biogeochemical consequences that have not as yet been fully explored but are
important to understand, especially when receiving waters downstream are
designated as Nitrate Vulnerable Zones, or where downstream transitional and near-
coastal waters are impacted.

**Data Availability**
Data are available to download from the NERC Environmental Information Data
Centre (see links provided in Heppell et al., 2016a, 2016b). DTC data are available
under an Open Government Licence from https://data.gov.uk/dataset/demonstration-
test-catchments-data-archive.






## Acknowledgements

This work was funded by the NERC Macronutrient Cycles thematic programme
(grant numbers NE/J012106/1 and NE/J011738/1) and by the Demonstration Test
Catchment phase 1 project (grant number WQ0211) co-funded by Defra,
Environment Agency (EA) and the Welsh Assembly Government (WAG). CMH and
AB would like to thank the team who helped to set up the Macronutrient Cycles
sampling network and collected and processed the water samples reported in this
study: John Keery and James Tempest. The DTC phase 1 sampling network was
installed by Robin Hodgkinson and Jeff Short and serviced by Fiona Grant and Carla
Richmond (all ADAS), while all laboratory analyses were completed by Geoff
Warren, Moragh Stirling and Chris Yates at the University of Reading. Geoff
Hardwicke of the Environment Agency, Blandford, kindly provided access to the
discharge data for Brixton Deverill on the River Wylye. We are also very grateful to
Natalie Ludgate and Angela Fells for analysing the water samples in our
laboratories. Finally, this work would not have been possible without the kind and
continual support of the landowners and tenant farmers at our six sites.





Table 1 Hydrological characteristics of the six sub-catchments in the Hampshire
Avon.

| Site code | Major geology | River | Stream order[a] | Catchment size (km²) | BFI[b] | BFIHOST[c] | Major land use[d] |
|---|---|---|---|---|---|---|---|
| AP | Clay (>99%) | Sem | 1(73%) 2(18%) **3**(9%) | 4.9 | 0.207 | 0.234 | Arable (5%), Grassland (95%) |
| AS | Clay (74%) | Sem | 1(54%) 2(26%) **3**(20%) | 26.0 | 0.549 | 0.372 | Arable (10%), Grassland (90%) |
| GN | Greensand (52%) | Nadder | 1(58%) 2(39%) **3**(3%) | 34.6 | 0.781 | 0.695 | Arable (46%), Grassland (33%) |
| GA | Greensand (50%) | W Avon | 1(47%) 2(31%) **3**(22%) | 59.2 | 0.744 | 0.861 | Arable (25%), Grassland (50%) |
| CE | Chalk (96%) | Ebble | 1 (28%) **2** (72%) | 58.9 | 0.906 | 0.953 | Arable (55%), Grassland (32%) |
| CW | Chalk (80%) | Wylye | 1 (60%) **2** (40%) | 53.5 | 0.901 | 0.931 | Arable (50%), Grassland (35%) |

[a] Strahler stream order with % contribution of stream order to the network and stream order at site in
bold; [b] Baseflow Index calculated using discharge data collected from July 2013-2014; [c] Baseflow
Index calculated using the UK Hydrology of Soil Types (HOST) classification; [d] Major Land use based
on 2010 agcensus data




Table 2 Summary of (a) linear mixed effects model parameters; and (b) regression
statistics.

| Model | Nitrate or DOC ~ BFI + (1 + BFI\|Time) | |
|---|---|---|
| Response variable | Nitrate | DOC |
| AIC | 10752.7 | 10576.2 |
| Fitting method | ML | ML |
| | | |
| Random effects | | |
| Intercept (time) | 7117 | 89601 |
| BFI | 10341 | 70703 |
| Residual | 11558 | 19051 |
| | | |
| Fixed effects | | |
| Intercept | -59.98 | 1668.2 |
| Slope | 520.62(±17.96) | -1679.55(±30.58) |


| Dependent | Independent | Correlation coefficient | Coefficient of determination | Slope (SE) | Intercept |
|---|---|---|---|---|---|
| Nitrate (17 sites) | BFIHOST | 0.928 | 0.861*** | 535(47) | -45 |
| Nitrate (17 sites) | % arable | 0.839 | 0.704*** | 640(70) | 130 |





Table 3 A summary of regression statistics for the relationships between log-Nitrate,
log-DOC and log-Nitrate:DOC molar ratio by site with log-discharge.

| Site | Dependent | R | R2 | B (SE) |
|------|-----------|-----|-----|--------|
| CE | Log(Nitrate) | 0.263 | 0.069*** | 0.053 (0.014)*** |
|    | Log(DOC) | 0.466 | 0.217*** | 0.254 (0.036)*** |
|    | Log-(DOC:Nitrate) | 0.375 | 0.140*** | 0.199 (0.037)*** |
| GA | Log(Nitrate) | 0.742 | 0.550*** | 0.206 (0.014)*** |
|    | Log(DOC) | 0.606 | 0.368*** | 0.403 (0.041)*** |
|    | Log-(DOC:Nitrate) | 0.342 | 0.117 | 0.196 (0.042)*** |
| AS | Log(Nitrate) | 0.501 | 0.251*** | 0.361 (0.047)*** |
|    | Log(DOC) | 0.542 | 0.294*** | 0.245 (0.029)*** |
|    | Log-(DOC:Nitrate) | 0.176 | 0.031* | -0.110 (0.047)* |

***$p<0.0001$; *$p<0.05$





Table 4 Export of nitrate and DOC expressed as % of total annual load at each site;
and Mean DOC:Nitrate ratio (+/- SE) by season.

|  | Season | | | |
|---|---|---|---|---|
|  | Summer | Autumn | Winter | Spring |
| *Nitrate Seasonal Load (as % of annual)* | | | | |
| AS | 3 | 26 | 45 | 26 |
| GA | 5 | 12 | 66 | 16 |
| CE | 6 | 4 | 57 | 31 |
| *DOC Seasonal Load (as % of annual)* | | | | |
| AS | 6 | 11 | 55 | 27 |
| GA | 5 | 15 | 56 | 22 |
| CE | 4 | 2 | 64 | 28 |
| *DOC:Nitrate molar ratio* | | | | |
| AS | 14.20 (0.81) | 5.08 (0.64) | 7.05 (0.45) | 6.13 (0.43) |
| GA | 1.36 (0.14) | 1.47 (0.16) | 1.19 (0.05) | 1.69 (0.11) |
| CE | 0.261 (0.01) | 0.232 (0.03) | 0.356 (0.03) | 0.379 (0.03) |










**List of Tables**

**List of Tables**
Table 1 Hydrological characteristics of the six sub-catchments in the Hampshire
Avon.
Table 2 Summary of (a) linear mixed effects model parameters; and (b) regression
statistics.
Table 3 A summary of regression statistics for the relationships between log-Nitrate,
log-DOC and log-Nitrate:DOC molar ratio with log-discharge.
Table 4 Export of nitrate and DOC expressed as % of total annual load at each site;
and mean DOC:Nitrate ratio (+/- SE) by season.

**List of Figures**
Figure 1. Catchment map of the Hampshire Avon showing study sites and geology.
Grey lines indicate sub-catchment boundaries delineated by topography.
Figure 2. Soil Moisture Deficit (mm) and daily rainfall totals (mm) from June 2013 to
June 2015.
Figure 3. Relationship between (a) nitrate concentration and BFI; (b) DOC
concentration and BFI, and; (c) DOC:nitrate molar ratio and BFI for six sub-
catchments in the Hampshire Avon.
Figure 4. Relationship between nitrate concentration and BFI for this study and
Environment Agency Harmonised Monitoring Scheme sites upstream of Salisbury in
the Hampshire Avon (June 2013 - 2014).
Figure 5. Relationship between electrical conductivity and discharge for three sub-
catchments of contrasting geology in the Hampshire Avon (a) Chalk - CE; (b)
Greensand - GA; and (c) clay – AS (June 2013 – 2015). Inset box-whisker plots
illustrate seasonal variations in electrical conductivity for Greensand (5b) and clay
(5c) sites. 5(d) illustrates riparian head (mAOD) in relation to river discharge at Site
GA.
Figure 6. Inter-site comparison of the relationship between (a) nitrate concentration
and discharge; (b) DOC and discharge, and; (c) DOC:nitrate molar ratio and





discharge for three sub-catchments of contrasting geology in the Hampshire Avon:
Chalk - CE; Greensand - GA; and Clay – AS (June 2013 – 2015).

Figure 7.  Seasonal variations in the relationship between nitrate, DOC and
DOC:nitrate molar ratio with discharge for three sub-catchments of contrasting
geology in the Hampshire Avon. (a) Chalk – CE; (b) Greensand – GA; (c) Clay – AS
(d) Clay – AS.

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





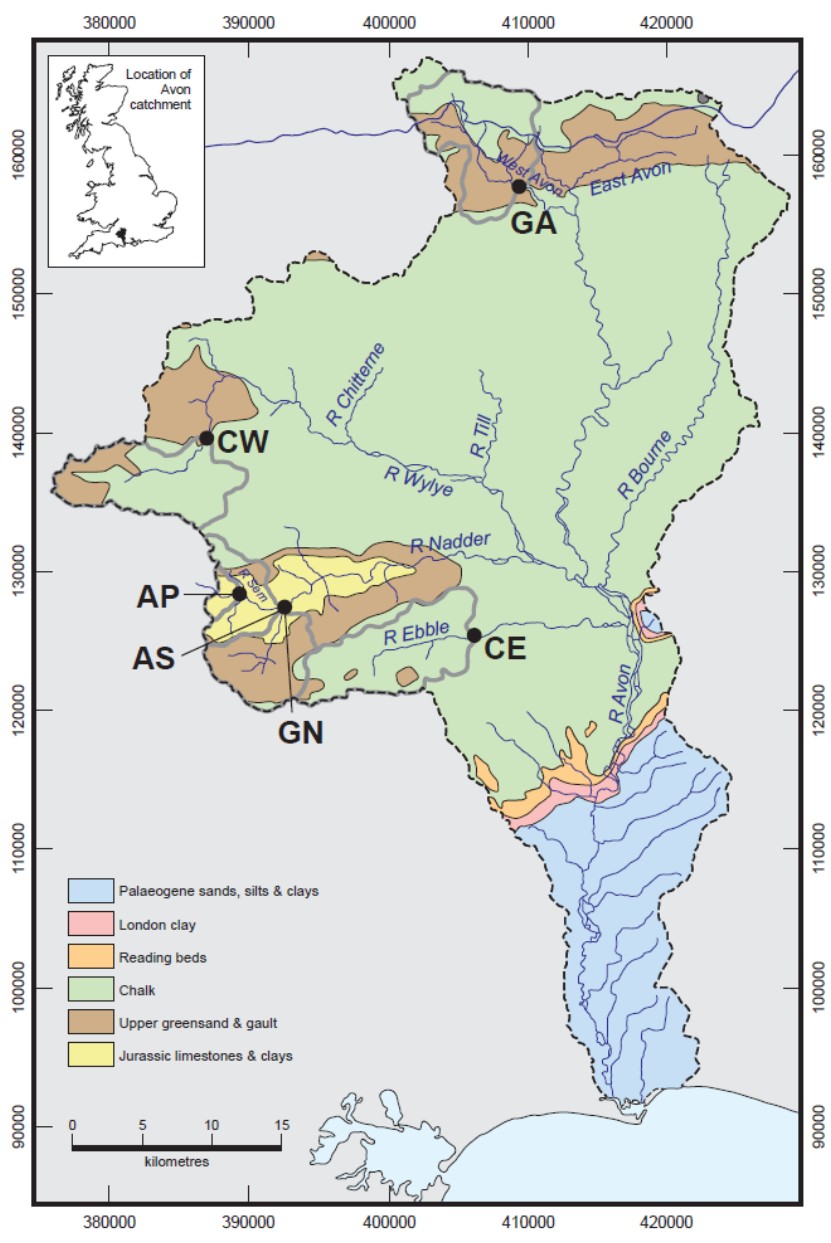



Figure 1. Catchment map of Hampshire Avon showing sites and geology. Grey lines
indicate sub-catchment boundaries delineated by topography.





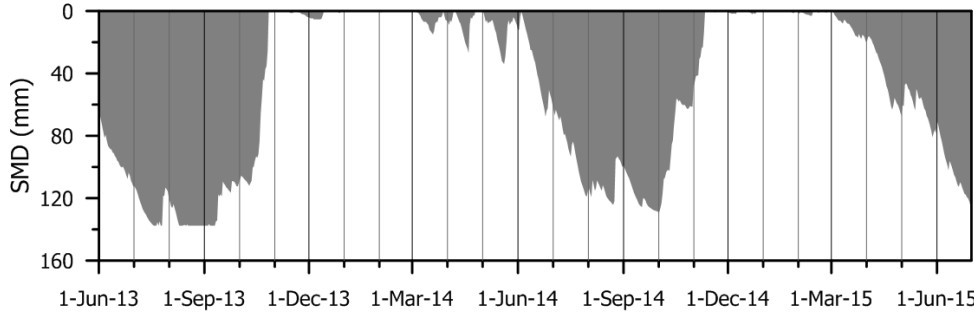

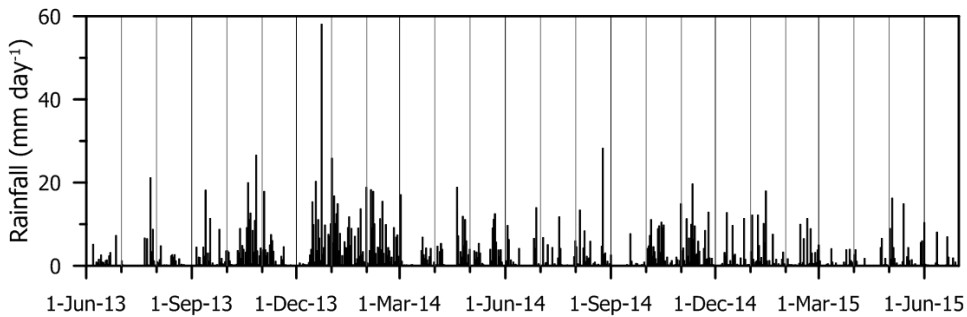

Figure 2. Soil Moisture Deficit (mm) and Daily Rainfall totals (mm) from June 2013 to June 2015.





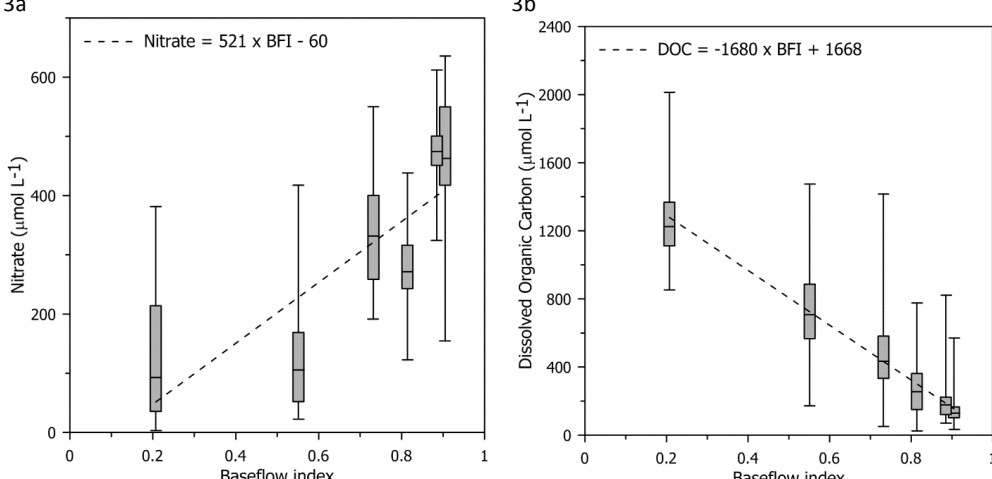

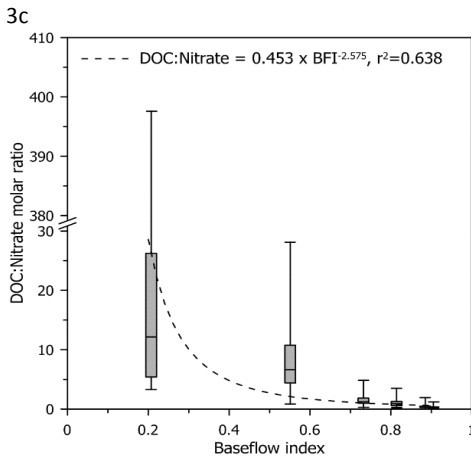


Figure 3. Relationship between (a) nitrate surface water concentration and baseflow
index; (b) DOC surface water concentration and baseflow index; and (c) DOC:nitrate
molar ratio and baseflow index for six sub-catchments in the Hampshire Avon.




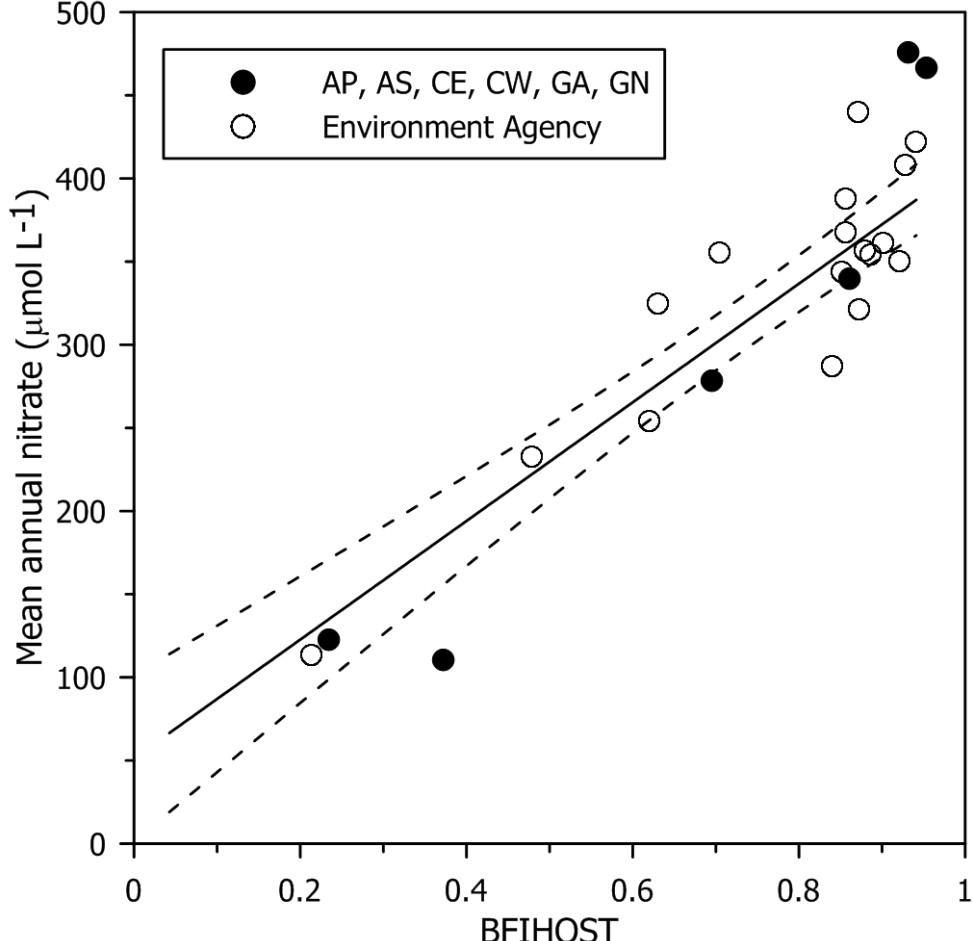


Figure 4. Relationship between nitrate concentration and baseflow index for this
study and Environment Agency Harmonised monitoring sites upstream of Salisbury
in the Hampshire Avon (June 2013 - 2014).






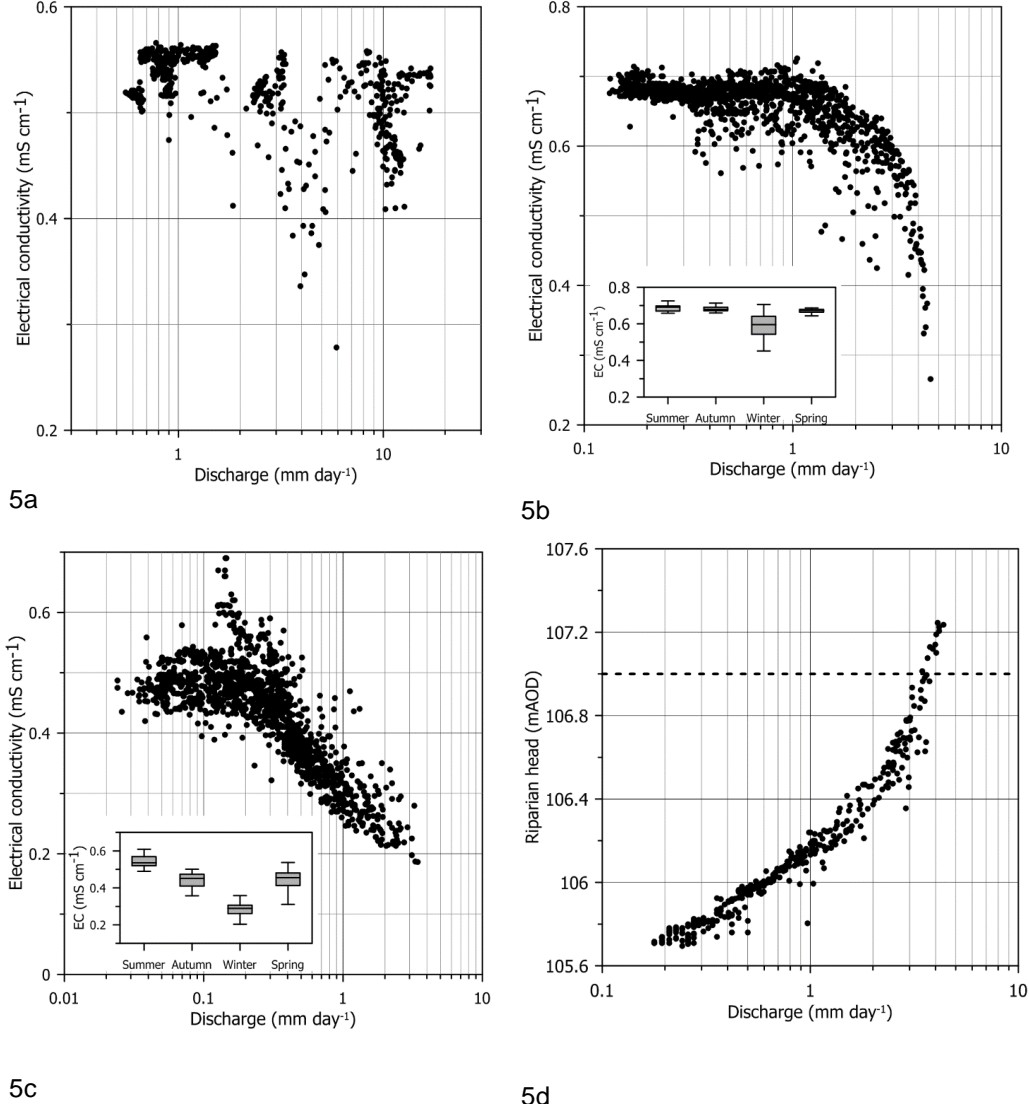

Figure 5. Relationship between Electrical Conductivity and Discharge for three sub-catchments of contrasting geology in the Hampshire Avon (a) Chalk - CE; (b) Greensand - GA; and (c) Clay – AS (June 2013 – 2015). Inset box-whisker plots indicate seasonal variations in electrical conductivity for Greensand (5b) and clay (5c) sites. 5(d) illustrates riparian head (mAOD) in relation to river discharge at Site GA.





1116

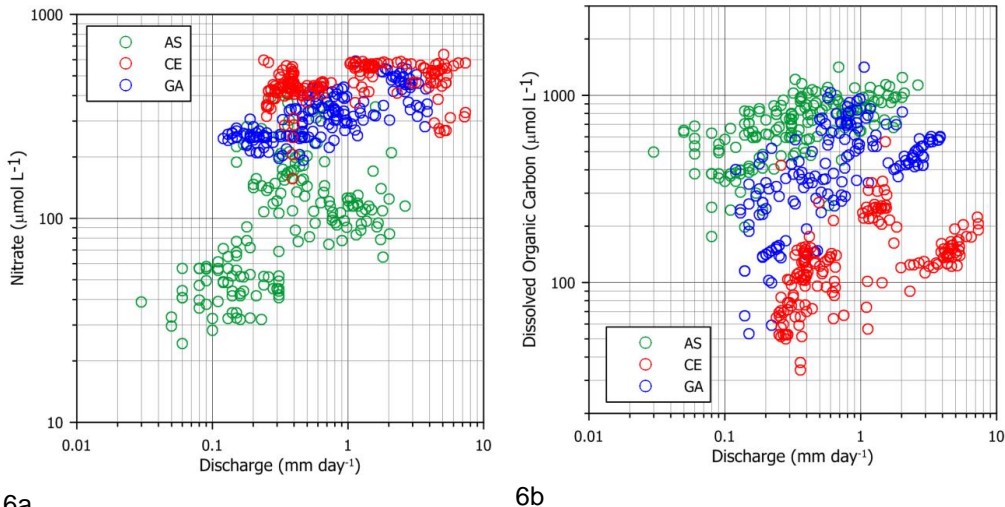

6a

6b

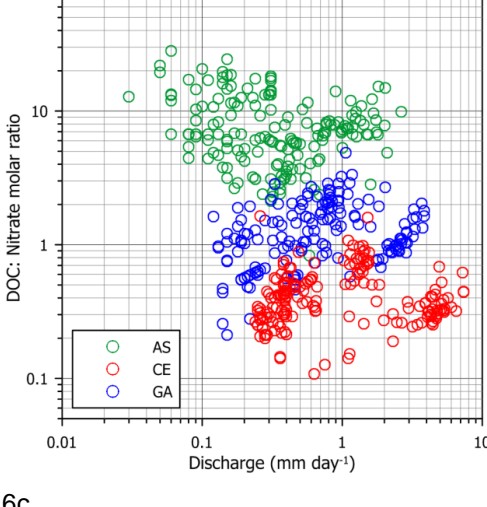

6c

1117

Figure 6. Inter-site comparison of the relationship between (a) nitrate concentration
and discharge; (b) DOC and discharge; and (c) DOC:nitrate molar ratio and
discharge for three sub-catchments of contrasting geology in the Hampshire Avon:
Chalk - CE; Greensand - GA; and Clay – AS (June 2013 – 2015).





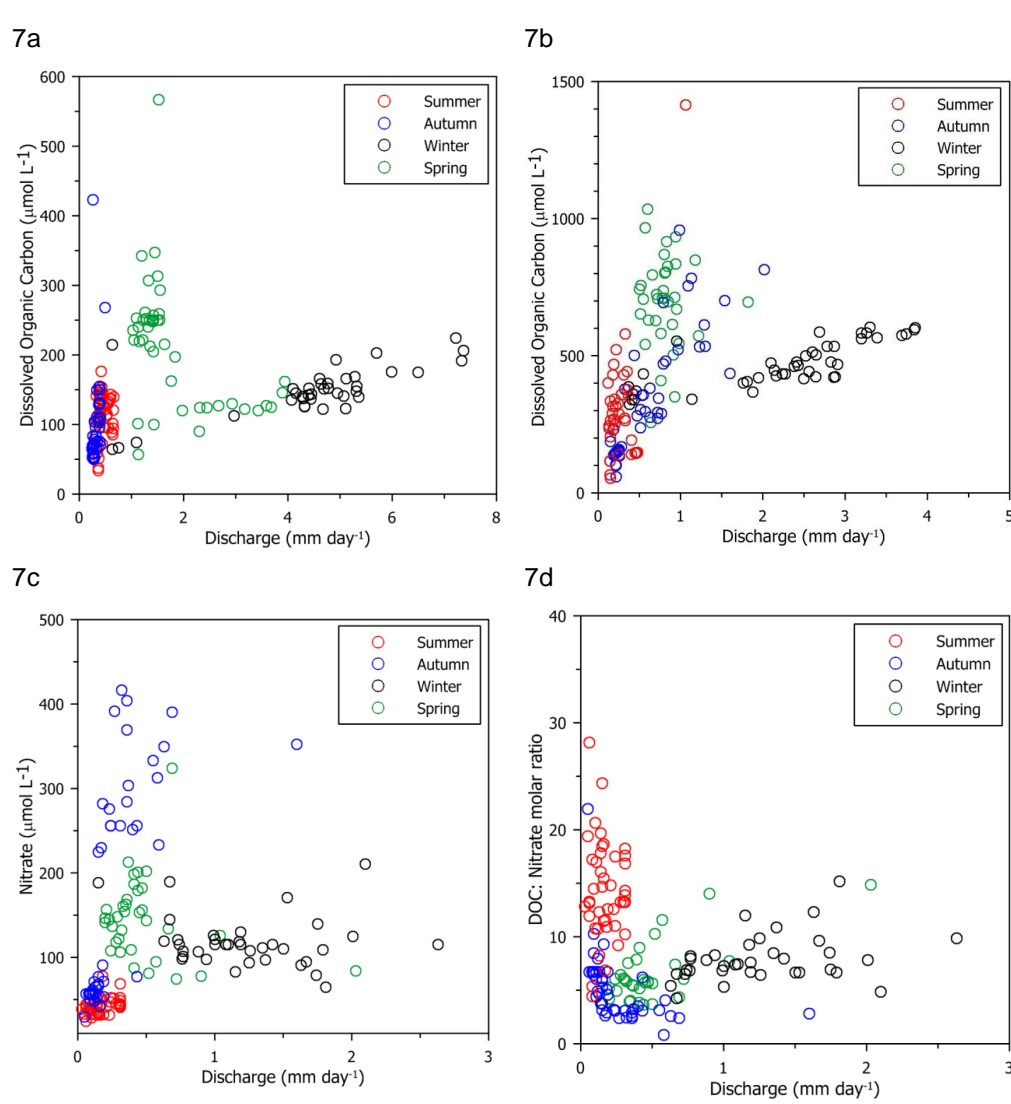


Figure 7. Seasonal variations in the relationship between solutes and discharge for
three sub-catchments of contrasting geology in the Hampshire Avon. (a) Chalk – CE;
(b) Greensand – GA; (c) Clay – AS (d) Clay – AS.
