# Peer review of "Hydrological controls on DOC:nitrate resource stoichiometry in a lowland, agricultural"

_Hydrology and Earth System Sciences, 2017_

## Referee Comment (RC1) · Anonymous Referee #1 · 12 Apr 2017

General comments: The paper fits very well to the multidisciplinary scope of HESS, connecting Hydrology, Ecology and Environmental questions. The data set is very interesting, because it covers more than a year with high temporal resolution and comprises an exceptional year which is predicted to become more frequent with climate change. This makes the results particularly interesting for management and predictions. Particularly the dynamics of both DOC and nitrate and their relationship is important in this context. The conclusions reached are relevant for nitrate management in agricultural catchments: Times of high nutrient load are defined for different hydrogeological sites in particularly varying with BFI. This data set and the approach is new to my knowledge. Over all the structure of the paper is logical and figures and

tables are appropriate. The discussion could be improved by picking up the points raised in the introduction and both could be more compact, for the reader to get your main points. I recommend publication after minor revisions. Please find some suggestions in the specific comments.

Specific comments:
Title: The title is appropriate
Abstract: Overall the abstract gives a good summary of the main findings, but the first and the last sentence could be improved:

$bullet$ L 22- L 26: This sentence is very long and confusing, so I would suggest breaking it into two. It is also unclear to me what role climate change (hydrology or DOC, nitrate production?) plays in this sentence. I suspect you refer to the reference of Whitehead et al., 2006 in the introduction. However, without the whole context this sentence is very confusing, as DOC and specially nitrate production and delivery arise from a variety of human impacts, whereas the impact of climate change on hydrology is well known to the reader when starting with the abstract.

$bullet$ L 42: The last sentence seems a bit disconnected here from the rest of the abstract, as suddenly DOC stands alone here. How about something like: Consequently, our study emphasizes the tight relationship between DOC availability and nitrate uptake in agricultural catchment and further reveals that this relationship is controlled to a great extent by the hydrological setting. Even though I agree with the authors that research from other catchments would be interesting to extrapolate the findings on a larger scale, I think that over all this is mentioned a bit too much throughout the paper e.g. what future work should do. I would appreciate a reduction of these sentences in the discussion too.

Introduction: The introduction comprises of 5 paragraphs, which cover (1) the need to study nitrate and the problem of managing nitrate in rivers, (2) the role of DOC in stream ecological processes, (3) the interplay of these two nutrients, (4) the specific situation in the UK and the predicted relationship with BFI and finally (5) the

hydrological controls on DOC:nitrate ratios.

*bullet* The paragraph 4 might be better integrated in paragraph 5, as it includes already predictions and goals (L 135-L 139). Therefore you might consider shifting this section to L 160. This way you would go from the DOC:nitrate removal, land use and climate change in L 125 from paragraph 3, directly to paragraph 5 starting with "Controls of riverine DOC and nitrate arise from…". After presenting these controls you could start explaining the specific situation of your study area and what you expect with BFI.

*bullet* L154-159: very long sentence, maybe a break at L 156: "…a wide range of BFI. We hypothesise…"

Methods: The methods are already very detailed; only the statistical part could be a bit more detailed. The linear mixed effect model approach seems appropriate to me. I just have a question, also concerning the way you report your results later: Could you please explain why you use two different R packages and different significance levels, as well as a different way of reporting them in your results? Chi2, F, r, r2,…

Results: Comprises of four subsections, which cover (1) Hydrological conditions, (2) BFI and nutrients, (3) BFI and (4) Seasonality: The titles of (2) and (3) could be a bit more specific. For example (2) "Quantification of the relationship between nutrients and BFI" and (3) "Intra-annual variations of groundwater and quickflow contribution"

*bullet* L 384 and L 386: Why are these results reported differently?

*bullet* L 467: It might be helpful to the reader to explain what your definition of old and new water is already at this point, even it is explained later in the discussion.

Discussion: The discussion has four subsections which do not follow the exactly same pattern as the goals stated in the introduction and continued in the results section. However, the order and the separation of the topics into the subsections, starting with hydrological aspects (1), continuing with BFI (2), then seasonality (3) and closing with environmental implications of this study (4) also seems logical to me. The discussion would benefit from the comparison with studies from other watersheds on DOC:nitrate molar ratios and hydrological responses, even if they are from other climate regions (maybe ones which are already characterized by hot and dry summers and wet

winters) or less agricultural areas (these are just some examples, but there are many others: Lupon, Anna, et al. "Contribution of pulses of soil nitrogen mineralization and nitrification to soil nitrogen availability in three Mediterranean forests." European Journal of Soil Science 67.3 (2016): 303-313; Sebestyen, Stephen D., Elizabeth W. Boyer, and James B. Shanley. "Responses of stream nitrate and DOC loadings to hydrological forcing and climate change in an upland forest of the northeastern United States." Journal of Geophysical Research: Biogeosciences 114.G2 (2009); Andrea, Butturini, et al. "Cross-site comparison of variability of DOC and nitrate c–q hysteresis during the autumn–winter period in three Mediterranean headwater streams: a synthetic approach." Biogeochemistry 77.3 (2006): 327-349. Tiemeyer, B., and P. Kahle. "Nitrogen and dissolved organic carbon (DOC) losses from an artificially drained grassland on organic soils." Biogeosciences 11.15 (2014): 4123.).

*bullet* L 496: Maybe you could introduce an abbreviation for EC and Q in the beginning and use it all the text, since both are used many times

*bullet* Section 4.3.: Here it would be useful if you could go back and pick up the points from your introduction, where you cite Whitehead et al., 2006 and Jiang et al., 2010 etc.: In the sense of does your study goes in line with their predictions and concerns?

*bullet* L 581: could arise from mineralisation? Please explain how exactly the DOC concentration can increase due to mineralisation. I could not find anything about this in Aubert et al., 2013 and to my knowledge mineralisation is a process that rather reduces DOC concentrations.

*bullet* L 662: A citation would be useful here to back up your statement

*bullet* L 677-682: This sentence is very long. Please make a point before and also suggests in L 679. In the second sentence you could say specifically winter, this way your conclusion becomes clearer. Conclusions: The conclusions could be a bit more to the point, meaning it is hard to understand from the conclusions, what are the main achievements of this study. Overall, I am wondering if the conclusions are really necessary, subsection 4.4. gives already a good idea on what the main findings and their implications are. If you keep the conclusions, I would suggest shorting them
to one paragraph. For example, L 688- L 690 is already explained in the discussion. Also L 707- L 711 could go out. L 714- L719: This sentence could be shorten: In this way, the spatial arrangement of areas of contrasting BFI within a catchment may have important ecological and biogeochemical consequences for receiving waters, especially if they are designated as NVZ or transitional and near-coastal areas.

Tables and Figures: In general tables and figures are clear and accompany well the text. I would suggest writing DOC instead of Dissolved Organic Carbon at the figure axes.

Technical comments: $bullet$ L 30, 31, 32 and 36: Baseflow Index is already defined as BFI in L 28

$bullet$ L 48: suggest to introduce abbreviation nitrogen (N) at this point

$bullet$ L 137: suggest to introduce abbreviation Baseflow Index (BFI) at this point

$bullet$ L 145: shouldn't the power function written like that with a small "a": C = aQb ?

$bullet$ As suggested already in the introduction: Baseflow Index -> BFI, capital letters or small? L 195 and L 196

$bullet$ L 206: you always wrote c., here you write circa

$bullet$ L 223: Temperature is written twice

$bullet$ L 266: Citation correct?

$bullet$ L304: Baseflow Index -> BFI

$bullet$ L328: Baseflow Index -> BFI

$bullet$ L 348: SMD already defined in Methods

$bullet$ L 371: round all $\chi2$ to 2 digits

$bullet$ L 388: There is a ) missing.

$bullet$ L 399: Citation Field, 2000 is 2002 in bibliography.

$bullet$ L 408: baseflow indices -> BFIs

$bullet$ L 469: occurs -> occur

$bullet$ L 475 and 479: just spring, without the?

$bullet$ L 599: Above the (instead of our) threshold. . .

$bullet$ L 650: greensand and chalk is written with capital letters throughout the paper.

[Figure]

Why different here?

*bullet* L 670: electrical conductivity (small letters)

*bullet* L 718: abbreviation NVZ already defined in discussion

*bullet* Figure 3: Suggest putting r2 and p in all figures, not necessary to write baseflow index everywhere, as it was already defined as BFI.

*bullet* Figure 5: Caption: in the text you write discharge and electrical conductivity always with small letters.

---

## Referee Comment (RC2) · Anonymous Referee #2 · 15 May 2017

General comments

This is an interesting and well-written paper that will be of interest for many in the hydrological community. The research questions investigated are relevant for managers of lowland agricultural catchments and the focus on DOC:Nitrate ratios is novel. The overall quality of the research is high, although occasionally there is a tendency to drift towards speculation (e.g. with reference to the different flowpaths operating in the catchments) without presenting all the necessary data to support these statements.

There are two key drawbacks to the paper at present that require addressing. The first is that nutrient samples were collected at 48 h intervals, which means that many short-term storm events may potentially have been missed and thus nutrient loads

underestimated. As no discharge time series are presented, it is not possible to say whether this is the case. The authors should recognise this explicitly in the text.

Secondly, the authors discuss at length how changes in C:N stoichiometry may limit the potential for in-stream uptake of inorganic nitrogen, but give far less attention to the equally important role of hydrological residence times in influencing these processes (e.g. Zarnetske et al, 2012, WRR). A better discussion of how these respective reaction and transportation factors interact to influence reach-scale nutrient transformation rates may be warranted here, particularly given the apparent importance of high discharge (and therefore velocity) periods for nutrient mobilisation and export.

Specific comments

L25: Climate change is undoubtedly important, but is not the only potential driver here. Land use change and management practices will also influence DOC and N dynamics. L39: 'Storm events' is perhaps somewhat misplaced in this context. Many readers will interpret this phrase as meaning short-term (hours) intense rainfall events that result in rapid changes in streamflow and biogeochemical dynamics. Yet the frequency of sampling in this study (48 h) is not sufficient to capture this variability. I suggest recasting this sentence (and similar others throughout the text) to clarify that 'storm events' relates more generally to the wetter conditions experienced during autumn and winter months. L92: I agree with the authors that more research into DOC dynamics in lowland agricultural streams is important. The Introduction as a whole is rather long and could be shortened considerably. As it stands, the key arguments do not stand out clearly. L127 and 139: It would be useful to state in this paragraph that baseflow index indicates the groundwater contribution to streamflow. Also, the authors should provide more justification for their prediction regarding the link between BFI and NO3. L141: This sentence seems repetitive of the start of the previous paragraph. L176: The start of Objective 3 seems repetitive of Objective 1. L225: Was the Manta 2 cleaned at any point during the study period? If so, at what frequency? Did this affect the results and if so, was a correction applied? L228: Can the authors confirm no sample

Interactive
comment

degradation occurred in the time between collection and analysis? Two weeks is a long time for samples to sit in an unpreserved state. L303: When comparing 48 hr nutrient samples with 15 min Q values, were instantaneous Q values at time of nutrient sampling used? Or were these integrated over longer time period? L381 and 465: Care is needed here to avoid placing text in the Results that would be better suited to the Discussion. L445: Are these trends descriptive only or can they be quantified? L462: A discharge time series would be nice to prove that 'autumn storms of intermediate discharge' are really storm events (as mentioned earlier) and not just seasonal shifts in baseflow. Throughout the Results section there are references to different years (e.g. L462) but this is not evident in the figures. L556: Some discussion of other land use types in the catchment and their potential influence on DOC would be useful here. Also, what potential is there for instream production? L612: Given that the discussion focuses heavily on flow pathways within the catchment, it would be helpful to show the rapid response of EC to rainfall events to support this statement. L664: This text could be expanded a little to place the results of this study in a wider context and make comparisons with previous research in this field.

Technical corrections

L127: Not just the UK. L135: Provide indicative range. L145: Define meaning of letters in equation. L154-160: Suggest splitting this very long sentence. L171: Clarify whether three or six sub-catchments are involved in the study. L195: Ref to support this? L215: Provide number of points and R2 value for stage-Q relationship. L266: How often were samples retrieved? L281: Provide precision and LOD information for autoanalyser and TOC-L. L291: Check reference date. L327: Provide indicative number of samples for those included in the analysis. L368: State type of correlation analysis used (Pearson or Spearman) L510: Does "the data" refer to EC-Q relationships? L571: Need to clarify here that the absolute concentration will change but the flow-weighted concentration won't (see Basu et al 2010 GRL) L573: By whom? Citation needed. Fig 1: Sites AS and GN seem in the same place. Also, can differences in baseflow indices be indicated

---

## Author Comment (AC1) · 18 Jun 2017

Authors responses to Reviewer #1

General comments: The paper fits very well to the multidisciplinary scope of HESS, connecting Hydrology, Ecology and Environmental questions. The data set is very interesting, because it covers more than a year with high temporal resolution and comprises an exceptional year which is predicted to become more frequent with climate change. This makes the results particularly interesting for management and predictions. Particularly the dynamics of both DOC and nitrate and their relationship is important in this context. The conclusions reached are relevant for nitrate management in

agricultural catchments: Times of high nutrient load are defined for different hydrogeo-logical sites in particularly varying with BFI. This data set and the approach is new to my knowledge. Over all the structure of the paper is logical and figures and tables are appropriate. The discussion could be improved by picking up the points raised in the introduction and both could be more compact, for the reader to get your main points. I recommend publication after minor revisions. Please find some suggestions in the specific comments.

We thank the reviewer for their positive comments in relation to scope, interest and relevance for management and future predictions in relation to climate change. Below we indicate the changes that we can make to address the reviewer's recommendations for improvements.

Abstract:

Overall the abstract gives a good summary of the main findings, but the first and the last sentence could be improved: bullet L 22- L 26: This sentence is very long and con-fusing, so I would suggest breaking it into two. It is also unclear to me what role climate change (hydrology or DOC, nitrate production?) plays in this sentence. I suspect you refer to the reference of Whitehead et al., 2006 in the introduction. However, without the whole context this sentence is very confusing, as DOC and specially nitrate produc-tion and delivery arise from a variety of human impacts, whereas the impact of climate change on hydrology is well known to the reader when starting with the abstract.

We can amend the sentence to as follows to address the reviewer's comments: 'The role that hydrology plays in governing the interactions between dissolved organic car-bon (DOC) and nitrogen in rivers draining lowland, agricultural landscapes is currently poorly understood. In light of the potential changes to the production and delivery of DOC and nitrate to rivers arising from climate change and land use management, there is a pressing need to improve our understanding of hydrological controls on DOC and nitrate dynamics in such catchments.'

bullet L 42: The last sentence seems a bit disconnected here from the rest of the abstract, as suddenly DOC stands alone here. How about something like: Consequently, our study emphasizes the tight relationship between DOC availability and nitrate uptake in agricultural catchment and further reveals that this relationship is controlled to a great extent by the hydrological setting. Even though I agree with the authors that research from other catchments would be interesting to extrapolate the findings on a larger scale, I think that over all this is mentioned a bit too much throughout the paper e.g. what future work should do. I would appreciate a reduction of these sentences in the discussion too.

We can remove the final sentence of the abstract and replace with the following text: 'Consequently, our study emphasizes the tight relationship between DOC and nitrate availability in agricultural catchments, and further reveals that this relationship is controlled to a great extent by the hydrological setting.'

Introduction:

The paragraph 4 might be better integrated in paragraph 5, as it includes already predictions and goals (L 135-L 139). Therefore you might consider shifting this section to L 160. This way you would go from the DOC:nitrate removal, land use and climate change in L 125 from paragraph 3, directly to paragraph 5 starting with "Controls of riverine DOC and nitrate arise from: : :". After presenting these controls you could start explaining the specific situation of your study area and what you expect with BFI.

We agree with the reviewer and can remove paragraph 4 from the introduction. We could adjust our research objective (i) to include a description of the contrasting geology that we are considering:

(i) To quantify the relationship between nitrate, DOC and DOC:nitrate molar ratio with Baseflow Index for six sub-catchments of contrasting geology (Chalk, Greensand and clay) in the Hampshire Avon.

L154-159: very long sentence, maybe a break at L 156: ": : :a wide range of BFI. We hypothesise: : :"

Agreed – we can make the suggested change.

Methods:

Methods: The methods are already very detailed; only the statistical part could be a bit more detailed. The linear mixed effect model approach seems appropriate to me. I just have a question, also concerning the way you report your results later: Could you please explain why you use two different R packages and different significance levels, as well as a different way of reporting them in your results? Chi2, F, r, r2,: : :

We use the lmer function in package lme4 for the surface water data, and the lme function in package nlme for the porewater data because the latter offered us the opportunity to compare mean nitrate and DOC porewater values between sites, whilst the former offered more flexibility with the mixed effect modelling. The two packages provide different statistical outputs. Bates et al (2015) provides a full description of the differences between the two packages. [Bates et al (2015) Journal of Statistical Software, doi 10.18637/jss.v067.i01].

Results:

Comprises of four subsections, which cover (1) Hydrological conditions, (2) BFI and nutrients, (3) BFI and (4) Seasonality: The titles of (2) and (3) could be a bit more specific. For example (2) "Quantification of the relationship between nutrients and BFI" and (3) "Intra-annual variations of groundwater and quickflow contribution" We can change the titles to those kindly suggested by the reviewer. bullet L 384 and L 386: Why are these results reported differently?

Please see comments above.

L 467: It might be helpful to the reader to explain what your definition of old and new water is already at this point, even it is explained later in the discussion.

We can add definitions to this text.

Discussion:

The discussion would benefit from the comparison with studies from other watersheds on DOC:nitrate molar ratios and hydrological responses, even if they are from other climate regions (maybe ones which are already characterized by hot and dry summers and wet winters) or less agricultural areas (these are just some examples, but there are many others:

Lupon, Anna, et al. "Contribution of pulses of soil nitrogen mineralization and nitrification to soil nitrogen availability in three Mediterranean forests." European Journal of Soil Science 67.3 (2016): 303-313

Sebestyen, Stephen D., Elizabeth W. Boyer, and James B. Shanley. "Responses of stream nitrate and DOC loadings to hydrological forcing and climate change in an upland forest of the northeastern United States." Journal of Geophysical Research: Biogeosciences 114.G2 (2009);

Andrea, Butturini, et al. "Cross-site comparison of variability of DOC and nitrate c–q hysteresis during the autumn–winter period in three Mediterranean headwater streams: a synthetic approach." Biogeochemistry 77.3 (2006): 327-349.

Tiemeyer, B.,and P. Kahle. "Nitrogen and dissolved organic carbon (DOC) losses from an artificially drained grassland on organic soils." Biogeosciences 11.15 (2014): 4123.).

We thank the reviewer for providing details of additional references to include in the paper. We do compare the results of our study with those obtained for multiple land use types across the USA, reported in the US LINX II study, but agree that more can be made of comparisons with other regions and land use types (although note that Lupon et al reference above does not provide DOC data to accompany the N data).

L 496: Maybe you could introduce an abbreviation for EC and Q in the beginning and use it all the text, since both are used many times

We find the use of too many abbreviations potentially confusing so would prefer to keep the text as is. In this case we have avoided using EC in particular due to potential confusion with our site CE.

Section 4.3.: Here it would be useful if you could go back and pick up the points from your introduction, where you cite Whitehead et al., 2006 and Jiang et al., 2010 etc.: In the sense of does your study goes in line with their predictions and concerns?

We could address this suggestion as follows:

Provided supplementary information (Figure S1) with a time series of discharge and nitrate for site AS to show the flushing of nitrate into the river during Autumn.

Add the following text to the manuscript:

'The elevated concentrations of nitrate observed in the River Sem in Autumn 2013 , provide some additional evidence to support results from dynamic modelling using INCA-N which show that drought conditions followed by wetting up of soil (as predicted in future climate change scenarios) can give rise to high nitrate loads in rivers (Whitehead et al., 2006). However, we observed this flushing effect most markedly in the clay sub-catchment of the Hampshire Avon where the majority of nitrate is likely to be delivered to the stream through shallow subsurface pathways, as opposed to the Chalk catchments where groundwater contributions of nitrate dominate.'

L 581: could arise from mineralisation? Please explain how exactly the DOC concentration can increase due to mineralisation. I could not find anything about this in Aubert et al., 2013 and to my knowledge mineralisation is a process that rather reduces DOC concentrations.

Yes rather than mineralisation we mean production of DOC. We apologise for the error. Aubert et al note the production of DOC follows inter-annual patterns controlled by surface biological processes influenced by temperature. We can alter the text accordingly.

L 662: A citation would be useful here to back up your statement

We propose to add Rodriguez-Cardona et al (2015) here to support the statement. L 677-682: This sentence is very long. Please make a point before and also suggests in L 679. In the second sentence you could say specifically winter, this way your conclusion becomes clearer.

Many thanks for the suggestion. We can amend the text as follows:

'Our research gives added impetus to the need to control autumn run-off from drained, grassland catchments supporting intensive livestock farming. Our study also suggests that during winter, periods of lateral flow and over-bank flooding in areas of intermediate BFI, such as Greensand, may export a significant proportion of the annual nitrate load with little opportunity for in-stream nitrate processing or removal.'

Conclusions:

The conclusions could be a bit more to the point, meaning it is hard to understand from the conclusions, what are the main achievements of this study. Overall, I am wondering if the conclusions are really necessary, subsection 4.4. gives already a good idea on what the main findings and their implications are. If you keep the conclusions, I would suggest shorting them to one paragraph.

For example, L 688- L 690 is already explained in the discussion. Also L 707- L 711 could go out.

We would like to keep the conclusion to highlight the main achievements of the study but we can shorten these to bullet points as suggested by the reviewer.

L 714- L719: This sentence could be shorten: In this way, the spatial arrangement of areas of contrasting BFI within a catchment may have important ecological and biogeochemical consequences for receiving waters, especially if they are designated as NVZ or transitional and near-coastal areas.

We can shorten the sentence following your helpful suggestion.

Tables and Figures: In general tables and figures are clear and accompany well the text. I would suggest writing DOC instead of Dissolved Organic Carbon at the figure axes.

We can use DOC instead of Dissolved Organic Carbon for Figure axes.

Please also note the supplement to this comment:
http://www.hydrol-earth-syst-sci-discuss.net/hess-2017-30/hess-2017-30-AC1-supplement.pdf

[Figure]

[Figure]

[Figure]

**Fig. 1.**

---

## Author Comment (AC2) · 18 Jun 2017

Authors responses to Reviewer #2

This is an interesting and well-written paper that will be of interest for many in the hydrological community. The research questions investigated are relevant for managers of lowland agricultural catchments and the focus on DOC:Nitrate ratios is novel. The overall quality of the research is high, although occasionally there is a tendency to drift towards speculation (e.g. with reference to the different flowpaths operating in the catchments) without presenting all the necessary data to support these statements. There are two key drawbacks to the paper at present that require addressing. The

first is that nutrient samples were collected at 48 h intervals, which means that many short-term storm events may potentially have been missed and thus nutrient loads underestimated. As no discharge time series are presented, it is not possible to say whether this is the case. The authors should recognise this explicitly in the text.

We propose to add a time series of discharge and nitrate as supplementary material. This will also help to address comments of Reviewer 1 in relation to autumn first flushing of nitrate. This will enable us to explicitly state that nutrient loads may be under-estimated due to frequency of monitoring (although note that we do not present absolute loads in this paper, but focus instead on proportional loads by season).

Secondly, the authors discuss at length how changes in C:N stoichiometry may limit the potential for in-stream uptake of inorganic nitrogen, but give far less attention to the equally important role of hydrological residence times in influencing these processes (e.g. Zarnetske et al, 2012, WRR). A better discussion of how these respective reaction and transportation factors interact to influence reach-scale nutrient transformation rates may be warranted here, particularly given the apparent importance of high discharge (and therefore velocity) periods for nutrient mobilisation and export.

The interplay between residence time and reaction rates in the hyporheic zone are the focus of another publication arising from this research in which we explicitly measure residence time and compare to reaction rate (but under baseflow conditions only). We would prefer to keep the focus here on the interactions between DOC and nitrate that were observed in the river and riparian porewater, because we do not quantify either reaction or transport here (i.e. we did not directly measure residence time or denitrification in the wider catchment so we can only speculate that autumn flushing of nitrate occurs in part due to reduced residence time in soils as water flows from the soil surface through macropores and artificial drains).

Abstract:

L39: 'Storm events' is perhaps somewhat misplaced in this context. Many readers

will interpret this phrase as meaning short-term (hours) intense rainfall events that result in rapid changes in streamflow and biogeochemical dynamics. Yet the frequency of sampling in this study (48 h) is not sufficient to capture this variability. I suggest recasting this sentence (and similar others throughout the text) to clarify that 'storm events' relates more generally to the wetter conditions experienced during autumn and winter months.

The relationship between Q and nitrate and/or DOC concentration during the winter period does capture the influence of storm events, but we also appreciate that sampling of finer, temporal resolution would also be beneficial to fully characterise storm events. We agree that in the abstract the emphasis should be on the winter and autumn periods rather than storm events per se, so we can amend the abstract by removing reference to storm events when describing responses of sub-catchments as follows: '(e.g. winter in sub-catchments underlain by Chalk and Greensand, and autumn in drained, clay sub-catchments)'

Introduction: L92: I agree with the authors that more research into DOC dynamics in lowland agricultural streams is important.

Thank you for the positive comment. We also feel that this area is currently under-researched, and yet potentially extremely important for nutrient cycling in such catchments.

The Introduction as a whole is rather long and could be shortened considerably. As it stands, the key arguments do not stand out clearly.

L127 and 139: It would be useful to state in this paragraph that baseflow index indicates the groundwater contribution to streamflow. Also, the authors should provide more justification for their prediction regarding the link between BFI and NO3.

We can remove this paragraph (following recommendation of Reviewer 1) to help shorten the introduction. We can also amend a later sentence to clarify that baseflow

index indicates the groundwater contribution to streamflow (see line 143-144).

'We might hypothesise that groundwater dominated areas (characterised by a high Baseflow Index). . . .'

L141: This sentence seems repetitive of the start of the previous paragraph.

We hope that by removing the previous paragraph we have addressed this redundancy.

L176: The start of Objective 3 seems repetitive of Objective 1.

We can shorten objective 3 to read as follows:

'To assess the potential implications of any spatio-temporal variations in DOC:nitrate ratios for future nitrogen management.' Methods:

L225: Was the Manta 2 cleaned at any point during the study period? If so, at what frequency? Did this affect the results and if so, was a correction applied?

Yes, all probes on the Manta were cleaned every two weeks, and a manual dip taken in the stilling well for calibration of the water level sensor. A correction was applied to the water level data when necessary on the fortnightly basis. The electrical conductivity probe was calibrated once a month. No drift in electrical conductivity was observable during the study.

L228: Can the authors confirm no sample degradation occurred in the time between collection and analysis? Two weeks is a long time for samples to sit in an unpreserved state.

We carried out laboratory experiments at the beginning of the project to check for evidence of sample degradation over the timescale of one month and did not find any significant sample degradation for nitrate and DOC over two weeks. In addition, at the time of each sample collection, a standard sample was left in the field (in the autosampler box) for the following 2 weeks to check for changes in solute concentration so that any necessary corrections could be applied. In practice this did not prove necessary

for nitrate or DOC.

L303: When comparing 48 hr nutrient samples with 15 min Q values, were instantaneous Q values at time of nutrient sampling used? Or were these integrated over longer time period?

Water samples were collected at 09:00 GMT every 2 days, so the corresponding Q value at 09:00 GMT on the same day of sample collection was used (i.e. instantaneous Q values at the time of nutrient sampling).

L381 and 465: Care is needed here to avoid placing text in the Results that would be better suited to the Discussion.

We feel that these sentences are largely descriptive observations and do not stray into the territory of a discussion so we would propose to leave these as they are.

Discussion:

L445: Are these trends descriptive only or can they be quantified?

We feel that the trends here are evident on the graph and do not require further quantification. We could calculate a slope for the first trend and a seasonal average for the second, but these would be site-specific.

L462: A discharge time series would be nice to prove that 'autumn storms of intermediate discharge' are really storm events (as mentioned earlier) and not just seasonal shifts in baseflow.

Many thanks for the useful comments in relation to storm events vs seasonal shifts in baseflow. In earlier drafts of the manuscript we considered including a time series but decided to keep the number of Figures manageable. In this revised version, following your recommendation, we offer supplementary material (Figure S2) that illustrates a time series for rainfall, electrical conductivity and discharge for the clay site, AS. We hope that this will address your comment above and also your request to show the

rapid response of electrical conductivity to rainfall to support the discussion later in the paper.

Throughout the Results section there are references to different years (e.g. L462) but this is not evident in the figures.

The Figure captions state the year of data collection (June 2013-2014 for the nitrate and DOC datasets).This information was missing from Figure 7 so we can add the dates.

L556: Some discussion of other land use types in the catchment and their potential influence on DOC would be useful here. Also, what potential is there for instream production?

The research took place in headwater catchments all selected for their rural characteristics (i.e. without significant contributions from urban land use) and without significant contributions from woodland. Major land use types (arable and grassland) in each catchment are summarised in Table 1 and we can add a statement as to the potential role of grassland in contributing DOC here. Elevated concentrations of DOC were observed in Spring and could potentially be due to in-stream production – this is explained in Line 580.

L612: Given that the discussion focuses heavily on flow pathways within the catchment, it would be helpful to show the rapid response of EC to rainfall events to support this statement.

Many thanks for this helpful suggestion. We have provided supplementary material (see Figure S2) with an example storm hydrograph to show the rapid response of EC to rainfall.

L664: This text could be expanded a little to place the results of this study in a wider context and make comparisons with previous research in this field.

Reviewer 1 also requested that the text be expanded in relation to references provided

in the Introduction and so we have suggested a way forward in our response to Reviewer 1. We agree that we can expand the text to make comparisons with previous research in this field.

Technical corrections:

L127: Not just the UK. L135: Provide indicative range.

This text will be removed following recommendation of Reviewer 1.

L145: Define meaning of letters in equation.

Definition of letters can be included in the text.

L154-160: Suggest splitting this very long sentence. Will do.

L171: Clarify whether three or six sub-catchments are involved in the study.

Six sub-catchments are included for the comparison with BFI and then a sub-set of three are used to illustrate seasonal trends.

L195: Ref to support this?

Geology is covered by Bristow et al 1999 reference above.

L215: Provide number of points and R2 value for stage-Q relationship.

C. 14 points were used for each site for the stage-Q relationship with r2 > 0.96

L266: How often were samples retrieved?

Samples were collected weekly. We will amend the text to state this.

L281: Provide precision and LOD information for autoanalyser and TOC-L.

For the inorganic nutrient analyses, and for DOC, we required an RSD of < 2%. Limits of detection for the analyses were 0.01 mg N/l NO3-N and 0.03 mg C/l DOC.

L291: Check reference date.

Should be corrected to 1992 in the text - thank you for pointing this out.

L327: Provide indicative number of samples for those included in the analysis.

Number of samples ranged from 12 to 56 depending on the site. Text can be added to state this.

L368: State type of correlation analysis used (Pearson or Spearman)

Pearson correlation analysis was used to explore relationships between solutes (nitrate and DOC) and BFI. We can add a statement in the data analysis section (3.2) to explain this.

L510: Does "the data" refer to EC-Q relationships?

Yes – we can clarify this in the text.

L571: Need to clarify here that the absolute concentration will change but the flow-weighted concentration won't (see Basu et al 2010 GRL)

Here we mean that the absolute concentration of geogenic solutes is maintained at higher discharge due to supply from the groundwater. We have not assessed flow-weighted concentration as we do not use data from multiple years as in Basu et al (2010); instead, we use the definition of chemostatic used in Godsey et al (2009). We can alter the text to clarify this as follows:

'The Chalk site (CE) is near-chemostatic with respect to total dissolved solutes and nitrate. This means that the concentration of geogenic solutes and nitrate is maintained at higher discharge, so that discharge drives solute load and hence the export of solutes to the coast. Here, we use the definition of near-chemostatic expressed in Godsey et al. (2009) as a slope of close to zero on a log(C)-log(Q) plot where C is concentration and Q is discharge.'

L573: By whom? Citation needed.

We will add Basu et al (2010) here.

Fig 1: Sites AS and GN seem in the same place. Also, can differences in baseflow indices be indicated.

AS and GN are very close together. They are at the confluence of two tributaries and the rivers run either side of the same field at this location so it is difficult to show the difference on a map of this scale. Baseflow index is summarised in Table 1 and we would prefer not to over-clutter the map with too much information.

Please also note the supplement to this comment:
http://www.hydrol-earth-syst-sci-discuss.net/hess-2017-30/hess-2017-30-AC2-supplement.pdf

[Figure]

**Fig. 1.**

[Figure]

**Fig. 2.**